# Optimal Approximation - Smoothness Tradeoffs for Soft-Max Functions

**Alessandro Epasto**
Google Research
aepasto@google.com

**Mohammad Mahdian**
Google Research
mahdian@google.com

**Vahab Mirrokni**
Google Research
mirrokni@google.com

**Manolis Zampetakis**
MIT
mzampet@mit.edu

## Abstract

A soft-max function has two main efficiency measures: (1) *approximation* - which corresponds to how well it approximates the maximum function, (2) *smoothness* - which shows how sensitive it is to changes of its input. Our goal is to identify the optimal approximation-smoothness tradeoffs for different measures of approximation and smoothness. This leads to novel soft-max functions, each of which is optimal for a different application. The most commonly used soft-max function, called exponential mechanism, has optimal tradeoff between approximation measured in terms of *expected additive approximation* and smoothness measured with respect to *Rényi Divergence*. We introduce a soft-max function, called *piecewise linear soft-max*, with optimal tradeoff between approximation, measured in terms of *worst-case* additive approximation and smoothness, measured with respect to $\ell_q$-*norm*. The worst-case approximation guarantee of the piecewise linear mechanism enforces *sparsity* in the output of our soft-max function, a property that is known to be important in Machine Learning applications [14, 12] and is not satisfied by the exponential mechanism. Moreover, the $\ell_q$-smoothness is suitable for applications in Mechanism Design and Game Theory where the piecewise linear mechanism outperforms the exponential mechanism. Finally, we investigate another soft-max function, called *power mechanism*, with optimal tradeoff between expected *multiplicative* approximation and smoothness with respect to the Rényi Divergence, which provides improved theoretical and practical results in differentially private submodular optimization.

## 1 Introduction

A soft-max function is a mechanism for choosing one out of a number of options, given the value of each option. Such functions have applications in many areas of computer science and machine learning, such as deep learning (as the final layer of a neural network classifier) [3, 2, 8], reinforcement learning (as a method for selecting an action) [20], learning from mixtures of experts [11], differential privacy [6, 15], and mechanism design [15, 10]. The common requisite in these applications is for the soft-max function to pick an option with close-to-maximum value, while behaving smoothly as the input changes.

The soft-max function that has come to dominate these applications is the *exponential function*. Given $d$ options with values $x_1, x_2, \ldots, x_d$, the exponential mechanism picks $i$ with probability $\exp(\lambda x_i)/(\sum_{j=1}^{d} \exp(\lambda x_j))$ for a parameter $\lambda > 0$. This function has a long history: It has been proposed as a model in decision theory in 1959 by Luce [13], and has its roots in the Boltzman (also

known as Gibbs) distribution in statistical mechanics [1, 7]. There are, however, many other ways to smoothly pick an approximately maximal element from a list of values. This raises the question: is there a way to quantify the desirable properties of soft-max functions, and are there other soft-max functions that perform well under such criteria? If there are such functions, perhaps they can be added to our repertoire of soft-max functions and might prove suitable in some applications.

These questions are the subject of this paper. We explore the tradeoff between the approximation guarantee of a soft-max function and its smoothness. A soft-max function is δ-*approximate* if the expected value of the option it picks is at least the maximum value minus $\delta$. Stronger yet, a function is δ-*approximate in the worst case* if it never picks an option of value less than the maximum minus $\delta$. We capture the requirement of *smoothness* using the notion of Lipschitz continuity. A function is Lipschitz continuous if by changing its input by some amount $x$, its output changes by at most a multiple of $x$. This multiplier, known as the Lipschitz constant, is then a measure of the smoothness of the function. This notion requires a way to measure distances in the domain (the input space) and the range (the output space) of the function.

We will show that if the $p$-norm and the Rényi divergence are used to measure distances in the domain and the range, respectively, then the exponential mechanism achieves the lowest possible (to within a constant factor) Lipschitz constant among all δ-approximate soft-max functions. This Lipschitz constant is $O(\log(d)/\delta)$. The exponential function picks each option with a non-zero probability, and therefore cannot guarantee an approximation in the worst case. In fact, we will show that for these distance measures, there is no soft-max function with bounded Lipschitz constant that can guarantee an approximation in the worst case.

On the other hand, if we use $p$-norms to measure changes in both the input and the output, new possibilities open up. We construct a soft-max function (called PLSOFTMAX, for piecewise linear soft-max) that achieves a Lipschitz constant of $O(1/\delta)$ and is also δ-approximate *in the worst case*. This is an important property, as it guarantees that the output of the soft-max function is always as sparse as possible. Furthermore, we prove that even only requiring δ-approximation in expectation, no soft-max function can achieve a Lipschitz constant of $o(1/\delta)$ for these distance measures.

We also study several other properties we might want to require of a soft-max function. Most notably, what happens if instead of requiring an additive approximation guarantee, we require a multiplicative one? A simple way to construct a soft-max function satisfying this requirement is to apply soft-max functions with additive approximation (e.g., exponential or PLSOFTMAX) on the logarithm of the values. The resulting mechanisms (the power mechanism, and LOGPLSOFTMAX) are Lipschitz continuous, but with respect to a domain distance measure called log-Euclidean. Moreover, we show that with the standard $p$-norm distance as the domain distance measure, no soft-max function with bounded Lipschitz constant and multiplicative approximation guarantee exists.

Finally, we explore several applications of the new soft-max functions introduced in this paper. First, we show how the power mechanism can be used to improve existing results (using the exponential mechanism) on differentially private submodular maximization. Second, we use PLSOFTMAX to design improved incentive compatible mechanisms with worst-case guarantees. Finally, we discuss how PLSOFTMAX can be used as the final layer of deep neural networks in multiclass classification.

## 1.1 Related Work

A lot of work has been done in designing soft-max function that fit better to specific applications. In Deep Learning applications, the exponential mechanism does not allow to take advantage of the sparsity of the categorical targets during the training. Several methods have been proposed to take use of this sparsity. Hierarchical soft-max uses a heuristically defined hierarchical tree to define a soft-max function with only a few outputs [18, 16]. Another direction is the use of a spherically symmetric soft-max function together with a spherical class of loss functions that can be used to perform back-propagation step much more efficiently [21, 4]. Finally there has been a line of work that targets the design of soft-max functions whose output favors sparse distributions [14, 12].

## 2 Definitions and Preliminaries

The $(d-1)$-dimensional *unit simplex* (also known as the *probability simplex*) is the set of all vectors $(x_1, \ldots, x_d) \in \mathbb{R}^d$ satisfying $x_i \geq 0$ for all $i$ and $\sum_{i=1}^{d} x_i = 1$. In other words, each point

in the $(d-1)$-dimensional unit simplex, which we denote by $\Delta_{d-1}$, corresponds to a probability distribution over $d$ possible outcomes $1, \ldots, d$.

**Soft-max.** A $d$-dimensional *soft-max function* (sometimes called a soft-max mechanism) is a function $\boldsymbol{f} : \mathbb{R}^d \to \Delta_{d-1}$. Intuitively, this corresponds to a randomized mechanism for choosing one outcome out of $d$ possible outcomes. For any $\boldsymbol{x} \in \mathbb{R}^d$, the value $x_i$ denotes the value of the outcome $i$, and $f_i(\boldsymbol{x})$ is the probability that $f$ chooses this outcome. In parts of this paper, specifically when we discuss multiplicative approximations, we restrict the outcome values $x_i$ to be positive, i.e., we consider soft-max functions from $\mathbb{R}_+^d$ to $\Delta_{d-1}$.

**Lipschitz continuity.** The Lipschitz property is defined in terms of a distance measure $d_1$ over $\mathbb{R}^d$ (the domain) and a distance measure $d_2$ over $\Delta_{d-1}$ (the range). A distance measure over a set is a function that assigns a non-negative *distance* to every ordered pair of points in that set. We do not require symmetry or the triangle inequality. We say that a soft-max function $f$ is $(d_1, d_2)$-Lipschitz continuous if there is a constant $\beta > 0$ such that for every two points $\boldsymbol{x}, \boldsymbol{y} \in \mathbb{R}^d$, we have

$$d_2(\boldsymbol{f}(\boldsymbol{x}), \boldsymbol{f}(\boldsymbol{y})) \leq \beta \cdot d_1(\boldsymbol{x}, \boldsymbol{y}). \tag{2.1}$$

The smallest $\beta$ for which the above holds is the Lipschitz constant of $\boldsymbol{f}$ (with respect to $d_1$ and $d_2$).

$\ell_p$ **distance and Rényi divergence.** Two measures of distance that are used in this paper are the $p$-norm distance and the Rényi divergence. For $p \geq 1$, the $p$-norm distance (also called the $\ell_p$ distance) between two points $\boldsymbol{x}, \boldsymbol{y} \in \mathbb{R}^d$ is denoted by $\|\boldsymbol{x} - \boldsymbol{y}\|_p$, and is defined as $\|\boldsymbol{x} - \boldsymbol{y}\|_p = \left( \sum_{i=1}^d |x_i - y_i|^p \right)^{1/p}$. For any $\alpha > 1$ and points $\boldsymbol{x}, \boldsymbol{y} \in \Delta_{d-1}$, the Rényi divergence of order $\alpha$ between $\boldsymbol{x}$ and $\boldsymbol{y}$ is denoted by $D_\alpha(\boldsymbol{x}\|\boldsymbol{y})$ and is defined as $D_\alpha(x\|y) = \frac{1}{\alpha-1} \log \left( \sum_{i=1}^d \frac{x_i^\alpha}{y_i^{\alpha-1}} \right)$. This expression is undefined at $\alpha = 1$, but the limit as $\alpha \to 1$ can be written as $D_1(\boldsymbol{x}\|\boldsymbol{y}) = \sum_{i=1}^d x_i \log \frac{x_i}{y_i}$ and is known as the Kullback-Leibler (KL) divergence. Similarly, the Rényi divergence of order $\infty$ can be defined as the limit as $\alpha \to \infty$, which is $D_\infty(\boldsymbol{x}\|\boldsymbol{y}) = \log \max_i \frac{x_i}{y_i}$.

**Approximation.** For any $\delta \geq 0$, a soft-max function $\boldsymbol{f} : \mathbb{R}^d \mapsto \Delta_{d-1}$ is $\delta$-*approximate* if

$$\forall \boldsymbol{x} \in \mathbb{R}^d : \qquad \langle \boldsymbol{x}, \boldsymbol{f}(\boldsymbol{x}) \rangle \geq \max_i \{x_i\} - \delta. \tag{2.2}$$

Note that the inner product $\langle \boldsymbol{x}, \boldsymbol{f}(\boldsymbol{x}) \rangle$ is the expected value of the outcome picked by $\boldsymbol{f}$. The function $\boldsymbol{f}$ is $\delta$-*approximate in the worst case* if

$$\forall \boldsymbol{x} \in \mathbb{R}^d, \forall i : \qquad f_i(\boldsymbol{x}) > 0 \Rightarrow x_i \geq \max_i \{x_i\} - \delta. \tag{2.3}$$

## 3 The Exponential Mechanism

The exponential soft-max function, parameterized by a parameter $\lambda$ and denoted by $\textsc{Exp}^\lambda$, is defined as follows: for $\boldsymbol{x} \in \mathbb{R}^d$, $\textsc{Exp}^\lambda(\boldsymbol{x})$ is a vector whose $i$'th coordinate is $\exp(\lambda x_i) / \sum_{j=1}^d \exp(\lambda x_j)$.

This mechanism was proposed and analyzed by McSherry and Talwar [15] for its application in differential privacy and mechanism design. It is not hard to see that the differential privacy property they prove corresponds to $(\ell_p, D_\infty)$-Lipschitz continuity, and therefore their analysis, cast in our terminology, implies the following:

**Theorem 3.1** ([15]). *For any $\delta > 0$ and $p, \alpha \geq 1$, the soft-max function $\textsc{Exp}^\lambda$ with $\lambda = \log(d)/\delta$ satisfies the following: (1) it is $\delta$-approximate, and (2) it is $(\ell_p, D_\alpha)$-Lipschitz continuous with a Lipschitz constant of at most $2\lambda$.*

This leaves the following question: is there any other soft-max function that achieves a better Lipschitz constant? The following theorem gives a negative answer.

**Theorem 3.2.** *Let $p, \alpha \geq 1$, $\delta > 0$, $d \geq 4$ and $\boldsymbol{f} : \mathbb{R}^d \to \Delta_{d-1}$ be a $\delta$-approximate soft-max function satisfying $D_\alpha(\boldsymbol{f}(\boldsymbol{x})\|\boldsymbol{f}(\boldsymbol{y})) \leq c \|\boldsymbol{x} - \boldsymbol{y}\|_p$ for all $\boldsymbol{x}, \boldsymbol{y} \in \mathbb{R}^d$. Then it holds $c > \frac{\log d - 2}{4\delta}$.*

Also, since the exponential mechanism assigns a non-zero probability to any outcome, it is of course not $\delta$-approximate in the worst case. The following theorem shows that this is an unavoidable property of any $(\ell_p, D_\alpha)$-Lipschitz continuous functions.

**Theorem 3.3.** *For any $p, \alpha \geq 1$, $\delta > 0$, there is no soft-max function that is $(\ell_p, D_\alpha)$-Lipschitz continuous and $\delta$-approximate in the worst case.*

The proofs of the above theorems are presented in the full version of the paper.

## 4 PLSOFTMAX: A Soft-Max Function with Worst Case Guarantee

As we saw in the last section, the exponential mechanism is a $(\ell_p, D_\infty)$-Lipschitz function with the best possible Lipschitz constant among all $\delta$-approximate functions. Furthermore, a worst case approximation guarantee is not possible for such Lipschitz functions. In this section, we focus on $(\ell_p, \ell_q)$-Lipschitz functions which are the soft-max functions that are used in mechanism design and in machine learning setting. These functions exhibit a different picture: the exponential function is no longer the best function in this family. Instead, we construct a soft-max function that achieves the best (up to a constant factor) Lipschitz constant and at the same time provides a worst-case guarantee. This is the most technical result of the paper.

### 4.1 Construction of PLSOFTMAX

While the analysis of the properties of PLSOFTMAX and understanding the intuition behind its construction might be technically challenging, its actual description is rather concise and simple. In this Section we give a complete description of this soft-max function, and state our main result. Due to lack of space, the proofs are left to the full version of the paper.

PLSOFTMAX is a piecewise linear function, where each linear piece is defined using a carefully designed matrix. More precisely, for a given $\boldsymbol{x} \in \mathbb{R}^d$, consider a permutation $\pi$ of $\{1, \ldots, d\}$ that *sorts x*, i.e., $x_{\pi(1)} \geq x_{\pi(2)} \geq \cdots \geq x_{\pi(d)}$, and let $\boldsymbol{P}_\pi$ be the permutation matrix of $\pi$, i.e., the matrix with 1's at entries $(i, \pi(i))$ and zeros everywhere else. In other words, $\boldsymbol{P}_\pi$ is the matrix that, once multiplied by $x$, sorts it. Each "piece" of our piecewise linear function corresponds to all $x \in \mathbb{R}^d$ that have the same sorting permutation $\pi$. The function, on this piece, is defined by multiplying $\boldsymbol{x}$ by $\boldsymbol{P}_\pi$ (thereby sorting it), then applying a linear function defined through a carefully designed family of matrices $\boldsymbol{SM}_{(k,d)}$, and then applying the inverse matrix $\boldsymbol{P}_\pi^{-1}$ to move values back to their original index. The matrices $\boldsymbol{SM}_{(k,d)}$ at the heart of this construction are defined below.

**Definition 4.1** (SOFT-MAX MATRIX). The soft max matrix $\boldsymbol{SM}_{(k,d)} = (m_{ij}) \in \mathbb{R}^{d \times d}$ is defined as $m_{11} = (k-1)/k$, $m_{ii} = 1/i$ for all $i \in [2, k]$, $m_{i1} = -1/k$ for all $i \in [2, k]$, $m_{ij} = -1/(j(j-1))$ for all $i, j \in [2, k]$ with $j < i$, and $m_{ij} = 0$ otherwise (See the full version of the paper for a better illustration of this matrix). Also, the vector $\boldsymbol{u}^{(k)} \in \mathbb{R}^d$ is defined as $u_i^{(k)} = 1/k$ if $i \leq k$ and $u_i^{(k)} = 0$ otherwise.

We consider partitions where each piece contains all vectors with the same ordering of the coordinates. Namely, for a permutation $\pi \in S_d$ we define $R_\pi$ to be the set of vectors $\boldsymbol{x} \in \mathbb{R}^d$ such that $x_{\pi(1)} \geq x_{\pi(2)} \geq \cdots \geq x_{\pi(d)}$. Also, let $\boldsymbol{P}_\pi$ be the permutation matrix of $\pi \in S_d$.

**Definition 4.2.** (PLSOFTMAX) Let $\delta > 0$, and consider a vector $\boldsymbol{x} \in \mathbb{R}^d$ with a sorting permutation $\pi$ and the corresponding permutation matrix $\boldsymbol{P}_\pi$. Define $k_{\boldsymbol{x}}$ as the maximum $k \in [d]$ such that $x_{\pi(1)} - x_{\pi(k)} \leq \delta$. The soft-max function PLSOFTMAX$^\delta$ on $x$ is defined as follows.

$$\text{PLSOFTMAX}^\delta(\boldsymbol{x}) = \frac{1}{\delta} \cdot \boldsymbol{P}_\pi^{-1} \cdot \boldsymbol{SM}_{(k_{\boldsymbol{x}}, d)} \cdot \boldsymbol{P}_\pi \cdot \boldsymbol{x} + \boldsymbol{P}_\pi^{-1} \cdot u^{(k_{\boldsymbol{x}})}. \quad (4.1)$$

As defined, it is not even clear that PLSOFTMAX$^\delta$ is a valid soft-max function, i.e., that PLSOFTMAX$^\delta(x) \in \Delta_{d-1}$. This, as well as the following result, is proved in the full version of the paper.

**Theorem 4.3.** *Let $\delta > 0$, PLSOFTMAX$^\delta$ be the function defined in (4.1) and let $\boldsymbol{x} \in \mathbb{R}^d$, then*

1. *PLSOFTMAX$^\delta$ is $\delta$-approximate in the worst case.*
2. *For any $p, q \geq 1$, PLSOFTMAX$^\delta$ is $(\ell_p, \ell_q)$-Lipschitz continuous with a Lipschitz constant that is at most*

$$\frac{2}{\delta} \min\{p+1, \frac{q}{q-1}, \log d\}.$$

The proof of Theorem 4.3 is based on bounding the $(p, q)$-subordinate norm of the matrices $\boldsymbol{SM}_{(k,d)}$. This is a challenging task since even computing the $(p, q)$-subordinate norm is NP-hard [19, 9]. To circumvent this, we generalize a theorem of [5] for subordinate norms, which might be of independent interest.

## 4.2 Lower Bounds and Comparison with the Exponential Function

Theorem 4.3 shows that the $(\ell_p, \ell_q)$-Lipschitz constant of PLSOFTMAX is at most $O(1/\delta)$ when $p$ is bounded or when $q$ is bounded away from 1, but becomes $O(\log(d)/\delta)$ when $(p, q)$ gets close to $(\infty, 1)$. It is easy to see that no soft-max function can achieve a Lipschitz constant better than $O(1/\delta)$. The following theorem shows that even for $(p, q) = (\infty, 1)$, no soft-max function can beat the bound proved in Theorem 4.3 for PLSOFTMAX. The proofs of this theorem and the other theorems in this Section are deferred to the full version of the paper.

**Theorem 4.4.** *Let $c, \delta > 0$, and assume $f : \mathbb{R}^d \to \Delta_{d-1}$ is a soft-max function that is $\delta$-approximate and $(\ell_\infty, \ell_1)$-Lipschitz continuous with a Lipschitz constant of at most $c$. Then, $c = \Omega\left(\log d/\delta\right)$.*

It is not hard to prove that for every $\boldsymbol{x}, \boldsymbol{y}$, $\|\boldsymbol{x} - \boldsymbol{y}\|_1 \leq \mathrm{D}_\infty\left(\boldsymbol{x} \| \boldsymbol{y}\right)$. Therefore, since the exponential soft-max function $\mathrm{EXP}^\lambda$ for $\lambda = \log(d)/\delta$ is $(\ell_p, \mathrm{D}_\infty)$-Lipschitz continuous with a Lipschitz constant of at most $2\lambda$ (Theorem 3.1), it must also be $(\ell_p, \ell_1)$-Lipschitz with the same constant. The following theorem shows that this Lipschitz constant is at least $\frac{\lambda}{2}$.

**Theorem 4.5.** *The $(\ell_p, \ell_1)$-Lipschitz constant of the soft-max function $\mathrm{EXP}^\lambda$ is at least $\frac{\lambda}{2}$. Therefore, the $(\ell_p, \ell_1)$-Lipschitz constant of a $\delta$-approximate exponential soft-max function is at least $\frac{\log d}{2\delta}$.*

The combination of the above result and Theorem 4.3 shows that in terms of the $(\ell_p, \ell_1)$-Lipschitz constant, there is a gap of $\Theta(\log d)$ between the exponential function and PLSOFTMAX.

# 5 Other variants and desirable properties

In the previous sections, we studied the tradeoff between Lipschitz continuity of soft-max functions and their approximation quality, as quantified by the maximum additive gap between the (expected) value of the outcome picked and the maximum value. In this section, we look into variants of our definitions and other desirable properties that we might need to require from the soft-max function. Most importantly, is it possible to require a multiplicative notion of approximation?

## 5.1 Multiplicative approximation

For any $\delta \geq 0$, we call a soft-max function $\boldsymbol{f} : \mathbb{R}_+^d \mapsto \Delta_{d-1}$ is $\delta$-*multiplicative-approximate* if for every $\boldsymbol{x} \in \mathbb{R}_+^d$, we have $\langle \boldsymbol{x}, \boldsymbol{f}(\boldsymbol{x}) \rangle \geq (1 - \delta) \max_i \{x_i\}$. Similarly, we can define the notion of $\delta$-multiplicative-approximate in the worst case.[1] Such multiplicative notions of approximation are practically useful in settings where the scale of the input is unknown.

First, here is a simple observation: to get a soft-max function with a multiplicative approximation guarantee, it is enough to start with one with an additive guarantee and apply it to the logarithm of the input values. The resulting function will be Lipschitz continuous, but with respect to a different distance measure as defined below.

**Definition 5.1.** *For any $\boldsymbol{x} \in \mathbb{R}_+^d$, let $\log(\boldsymbol{x}) := (\log(x_1), \ldots, \log(x_d))$. For $p \geq 1$, the $p$-log-Euclidean distance between two points $\boldsymbol{x}, \boldsymbol{y} \in \mathbb{R}_+^d$ is denoted by $\mathrm{Log}\text{-}\ell_p(\boldsymbol{x}, \boldsymbol{y})$ and is defined as $\mathrm{Log}\text{-}\ell_p(\boldsymbol{x}, \boldsymbol{y}) := \ell_p(\log(\boldsymbol{x}), \log(\boldsymbol{y}))$. Note that $\mathrm{Log}\text{-}\ell_p$ is a metric.*

We can now state the above observation as follows:

**Proposition 5.2.** *Let $\boldsymbol{f} : \mathbb{R}^d \to \Delta_{d-1}$ be a soft-max function that is $\delta$-approximate and $(\ell_p, \chi)$-Lipschitz for a distance measure $\chi$. Then the function $\mathrm{Log}\boldsymbol{f} : \mathbb{R}_+{}^d \mapsto \Delta_{d-1}$ defined by $\mathrm{Log}\boldsymbol{f}(x) := \boldsymbol{f}(\log(x))$ is a $\delta$-multiplicative-approximate soft-max function that is $(\mathrm{Log}\text{-}\ell_p, \chi)$-Lipschitz with the same Lipschitz constant as $\boldsymbol{f}$.*

Applying this proposition to PLSOFTMAX, we obtain a soft-max function called LOGPLSOFTMAX that is $\delta$-multiplicative-approximate in the worst case and (Log-$\ell_p$, $\ell_q$)-Lipschitz. More notably, applying this proposition to the exponential function, we obtain a soft-max function that we call the *power mechanism*, with a very simple and natural description: The Power Mechanism $\text{POW}^\lambda$ with parameter $\lambda$, applied to the input vector $x \in \mathbb{R}^d_+$ is defined as $\text{POW}^\lambda_i(\boldsymbol{x}) = x_i^\lambda / \sum_{j=1}^d x_j^\lambda$. We will see in Section 6 how this mechanism can be used to improve existing results in a differentially private optimization problem.

A question that remains is whether, to obtain a multiplicative approximation, it is necessary to switch the domain distance measure to Log-$\ell_p$. In other words, are there $\delta$-multiplicative-approximate soft-max functions that are Lipschitz with respect to the domain metric $\ell_p$? The following theorem, whose proof is deferred to the appendix, provides a negative answer.

**Theorem 5.3.** *For $\delta > 0$, let $\boldsymbol{f} : \mathbb{R}^d \to \Delta_{d-1}$ be a $\delta$-multiplicative-approximate soft-max function. Then there is no $p, q$ such that $\boldsymbol{f}$ is $(\ell_p, \ell_q)$-Lipschitz with a bounded Lipschitz constant. Similarly, there is no $p, \alpha$ such that $f$ is $(\ell_p, D_\alpha)$-Lipschitz with a bounded Lipschitz constant.*

### 5.2 Scale and Translation Invariance

Related to the notion of multiplicative approximation, one might wonder if there are soft-max functions that are *scale invariant*, i.e., guarantee that for every $\boldsymbol{x} \in \mathbb{R}^d$ and $c \in \mathbb{R}$, $\boldsymbol{f}(c\boldsymbol{x}) = \boldsymbol{f}(\boldsymbol{x})$? Similarly, one may require *translation invariance*, i.e., that for every $\boldsymbol{x} \in \mathbb{R}^d$ and $c \in \mathbb{R}$, $\boldsymbol{f}(\boldsymbol{x} + c \cdot \boldsymbol{1}) = \boldsymbol{f}(\boldsymbol{x})$. It is easy to see that indeed the mechanisms EXP and POW are translation and scale invariant, respectively. It is less obvious, but still not difficult, to show that similarly, the mechanisms PLSOFTMAX and LOGPLSOFTMAX are translation and scale invariant, respectively.

In fact, it turns out that translation and scale invariance go hand-in-hand with the notion of approximation: no scale-invariant function can guarantee additive approximation, and no translation-invariant function can guarantee multiplicative approximation.

## 6 Applications

We present three applications of the soft-max functions introduced in this paper. In Section 6.1, we show how to use PLSOFTMAX to design approximately incentive compatible mechanisms. In Section 6.2 we use POW to improve a result on differentially private submodular maximization. Finally, in Section 6.3, we discuss potential applications of PLSOFTMAX in neural network classifiers.

### 6.1 Approximately Incentive Compatibile Mechanisms via PLSOFTMAX

Let us start with an abstract definition of incentive compatibility in mechanism design. Consider a setting with $n$ self-interested agents, indexed $1, \dots, n$. A mechanism is a (randomized) algorithm $A$ that must pick one of the possible outcomes in a set $\Omega$. For simplicity, let us assume that $\Omega$ is finite and $|\Omega| = d$. Each agent $i$ has a utility function $u_i \in \mathbb{R}^\Omega_+$ that specifies the value that $i$ places on each of the possible outcomes. Let $\mathcal{U} \subseteq \mathbb{R}^\Omega_+$ denote the space of all possible utility functions. The input of the algorithm $A$ is the reported utility of all the agents, i.e., $A$ takes a $u \in \mathcal{U}^n$ as input, and probabilistically picks an outcome $A(u)$ in $\Omega$. We say that $A$ is $\varepsilon$-*incentive compatible* with respect to $\mathcal{U}$ if for every $u \in \mathcal{U}^n$, $u' \in \mathcal{U}$ and every agent $i$, the following inequality holds $\mathbb{E}_{z \sim A(\boldsymbol{u})} [u_i(z)] \geq \mathbb{E}_{z \sim A(u', \boldsymbol{u}_{-i})} [u_i(z)] - \varepsilon$.

Typically, in mechanism design, the challenge is to design a mechanism $A$ that is incentive compatible and at the same time (approximately) optimizes a given objective function $w$ that depends on the utility of the agents $u \in \mathcal{U}^n$ as well as the selected outcome in $\Omega$. At a high-level, a soft-max function can be used to design an incentive compatible mechanism as follows: Assume $f : \mathbb{R}^d \to \Delta_{d-1}$ is $(\chi, \ell_1)$-Lipschitz with respect to some domain distance measure $\chi$. The mechanism $A_f$ is defined as follows: it computes the value of all outcomes in $\Omega$ at the reported utilities $u \in \mathcal{U}^n$, and uses $f$ to pick an outcome with respect to these values.

A central concept is the sensitivity of the function $w$ with respect to $\chi$. The $\chi$-sensitivity $S_\chi(w)$ of $w$ is defined as $S_\chi(w) = \max\{\chi(\boldsymbol{w}(\boldsymbol{v}), \boldsymbol{w}(\boldsymbol{v}_{-i}, v'_i))\}$, where $\boldsymbol{w}(\boldsymbol{v}) = (w(\boldsymbol{v}, 1), \dots, w(\boldsymbol{v}, d))$ and the maximum is taken over all possible $i, \boldsymbol{v}, v'_i$. If the soft-max function $f$ has low $(\chi, \ell_1)$-Lipschitz

constant, and the objective $w$ has low sensitivity with respect to $\chi$, we can use the following theorem to obtain an $\varepsilon$-incentive compatible mechanism.

**Theorem 6.1.** *Assume a mechanism design setting where utilities of the agents are bounded from above by 1, i.e., $\mathcal{U} \subseteq [0,1]^\Omega$. Let $f : \mathbb{R}^d \to \Delta_d$ be a soft-max function with $(\chi, \ell_1)$-Lipschitz constant at most $L$, and $w : \mathcal{U}^n \times \Omega \to \mathbb{R}$ be an objective function. The algorithm $A_f$ is $(L/S_\chi(w))$-incentive compatible with respect to $\mathcal{U}$.*

The connection between soft-max and mechanism design established in the above theorem is not new. McSherry and Talwar [15] originally pointed out this connection and used it to design incentive compatible mechanisms. However, they stated this connection in terms of differential privacy (closely related to the $(\chi, D_\infty)$-Lipschitz property). The main difference between the above theorem and the one by McSherry and Talwar is that we only require $(\chi, \ell_1)$-Lipschitz continuity, which is closer to what the application demands. This can be combined with the soft-max function PLSOFTMAX analyzed in Theorem 4.3 to obtain results that were not achievable using the exponential mechanism. We present two applications of this here. See the full version of the paper for details and proofs.

**Worst-Case Guarantees for Mechanism Design.** If we replace the exponential mechanism with PLSOFTMAX in many applications of Differential Privacy in Mechanism Design, we get approximate incentive compatible algorithms with *worst-case* approximation guarantees as opposed to the expected approximation or the high-probability guarantees that are currently known. Consider for example the digital goods auction problem from [15], where $n$ bidders have a private utility for a good at hand for which the auctioneer has an unlimited supply and let $\mathrm{R_{OPT}}$ be the optimal revenue that the auctioneer can extract for a given set of bids. We can then prove the following.

**Informal Theorem 6.2.** *There is an $\varepsilon$-incentive compatible mechanism for the digital goods auction problem where the revenue of the auctioneer is at least $\mathrm{R_{OPT}} - O(\mathrm{R_{OPT}} \cdot n)/\varepsilon)$ **in the worst-case**.*

**Better Sensitivity implies better Utility.** If the revenue objective function $w$ has bounded $S_q(w)$ sensitivity for some $q < \log(d)$, then using PLSOFTMAX we get a significantly better revenue-incentive compatibility tradeoff compared to using the exponential mechanism. This is clear form Lemma 6.1, and Theorems 4.3 and 3.1. See the full version of the paper for details.

## 6.2 Differentially Private Submodular Maximization via the Power Mechanism

We now show that $D_\infty$ smoothness can be used to design differentially private algorithms.

**Differential Privacy.** A randomized algorithm $A$ satisfies $\varepsilon$-*differential privacy* if $\mathbb{P}(A(\boldsymbol{v}) \in S) \leq \exp(\varepsilon) \cdot \mathbb{P}(A(v_i', \boldsymbol{v}_{-i}) \in S)$ for all $i \in [n]$, $\boldsymbol{v} \in \mathcal{D}^n$, $v_i' \in \mathcal{D}$ and all sets $S \subseteq \Omega$, where $\Omega$ is the set of possible outputs of $A$.

For some distance metric $\chi$ of $\mathbb{R}_+^d$, the soft-max function $\boldsymbol{f}$ satisfies $\mathrm{D_\infty}(\boldsymbol{f}(\boldsymbol{x}) \| \boldsymbol{f}(\boldsymbol{y})) \leq L \cdot \chi(\boldsymbol{x}, \boldsymbol{y})$ $\forall \boldsymbol{x}, \boldsymbol{y} \in \mathbb{R}_+^d$ and can be used to design differentially private algorithms when the objective function has low $\chi$ sensitivity, according to the following lemma. The proof of this Lemma follows directly from the definitions of $\mathrm{D_\infty}$ and $S_\chi$.

**Lemma 6.3.** *Let $\boldsymbol{f} : \mathbb{R}_+^d \to \Delta_d$ be a soft maximum function, $w : \mathcal{D}^n \to \mathbb{R}_+$ be an objective function. If $\boldsymbol{f}$ is $L$-Lipschitz with respect to $\mathrm{D_\infty}$ and $\chi$, then $A_{\boldsymbol{f}}$ is $(L/S_\chi(w))$-differentially private.*

### 6.2.1 Application to Differentially Private Submodular Optimization

In differentially private maximization of submodular functions under cardinality constraints, we observe that if the input data set satisfies a mild assumption, then using power mechanism we achieve an asymptotically smaller error compare to the state of the art algorithm of Mitrovic et al. [17].

**Submodular Functions.** Let $\mathcal{D}$ be a set of elements with $d = |\mathcal{D}|$. A function $h : 2^\mathcal{D} \to \mathbb{R}_+$ is called submodular if $h(R \cup \{v\}) - h(R) \geq h(T \cup \{v\}) - h(T)$ for all $R \subseteq T \subseteq \mathcal{D}$ and all $v \in \mathcal{D} \setminus T$.

**Monotone Functions.** A function $h : 2^\mathcal{D} \to \mathbb{R}_+$ is monotone if $h(T) \geq h(R)$ for all $R \subseteq T \subseteq \mathcal{D}$.

Monotone and Submodular Maximization under Cardinality Constraints is the optimization problem $\max_{R \subseteq \mathcal{D}, |R| \leq k} h(R)$. We use the Algorithm 1 of [17], where we replace the exponential mechanism

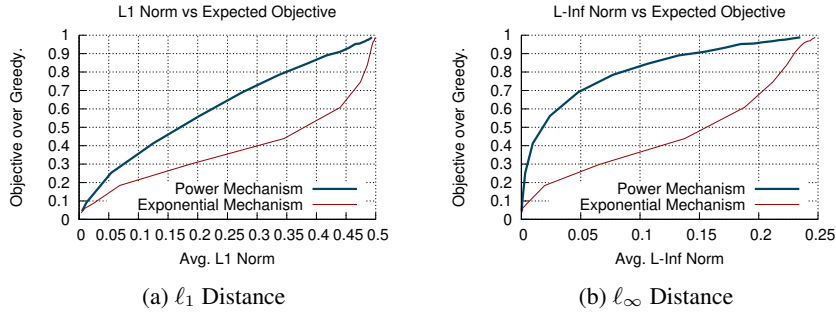

| (a) $\ell_1$ Distance | (b) $\ell_\infty$ Distance |

Figure 1: Smoothness vs utility in the submodular maximization with cardinality constraint $k = 10$. The y-axis shows the ratio of the average objective to the (non-private) greedy algorithm. The x-axis represents the sensitivity to the manipulation test of the value of the first element selected.

in the soft maximization step with the power mechanism. Let $S_{l,q}$ be the sensitivity of $h$ with respect to $q$-log-Euclidean quasi-metric and OPT be the optimal value.

**Theorem 6.4.** *Let $h : 2^{\mathcal{D}} \to \mathbb{R}_+$ be a monotone and submodular function. Then, there exists an efficient $(\varepsilon, \delta)-$differentially private algorithm with output $R_k$ that achieves multiplicative approximation guarantee* $\mathbb{E}[h(R_k)] \geq \left(1 - \exp\left(d^{\frac{\varepsilon}{S_{l,\infty}(h)\left(\sqrt{k}+\sqrt{\log(1/\delta)}\right)} - 1}\right)\right) \text{OPT}.$

Even though it is not immediately clear if the above guarantee is better than the one of [17], we note that the above result has only a multiplicative approximation error. In contrast, the algorithm of [17] has both multiplicative and additive error. In general, it is impossible to compare the two tradeoffs, because the tradeoff of [17] is parameterized by $S_\infty$ sensitivity of $h$ whereas our tradeoff is parameterized by $S_{l,\infty}$ of $h$. Even though there is no a priori comparison between the two sensitivities, the following mild assumption allows us to compare them.

**Definition 6.5** ($t$-MULTIPLICATIVE INSENSITIVITY)**.** *A data-set $\mathcal{D}^n$ is $t$-multiplicative insensitive for an objective function $w : \mathcal{D}^n \times [d] \to \mathbb{R}_+$ if for any two inputs $\boldsymbol{V}, \boldsymbol{V'} \in \mathcal{D}^n$ that differ only in one coordinate and for any $i \in [d]$, if $w(\boldsymbol{V'}, i) \leq w(\boldsymbol{V}, i)$ it holds that $\frac{w(\boldsymbol{V'},i)}{w(\boldsymbol{V},i)} \geq 1 - \frac{1}{t}\frac{S_\infty(w)}{\text{OPT}}$.*

Based on the above definition, we prove that the error of the power mechanism, under the assumption of $O(1)$-multiplicative insensitivity, is asymptotically better than the error of the exponential mechanism. This improvement is also observed in experiments with real world data as it is shown in Figure 1. The missing proofs and a detailed explanation of results is in the the full version of the paper.

**Corollary 6.6.** *Assume the input data satisfy $O(1)$-multiplicative insensitivity. Let $T_k$ be the output of Algorithm 1 of [17] using the exponential mechanism, then the approximation guarantee is*

$$\mathbb{E}[h(T_k)] \geq (1 - 1/e)\,\text{OPT} - O\left(k \cdot S_\infty(h)\log|\mathcal{D}|/\varepsilon\right)$$

*whereas if $R_k$ is the output when using the power mechanism, then the approximation guarantee is*

$$\mathbb{E}[h(R_k)] \geq (1 - 1/e)\,\text{OPT} - O\left(\sqrt{k} \cdot S_\infty(h)\log|\mathcal{D}|/\varepsilon\right).$$

We validated these theoretical results with an empirical study we report fully in the full version of the paper. Here we briefly outline our results in Figure 1, where we show improved objective vs sensitivity trade-offs for the power mechanism in an empirical data manipulation tests. In this experiments we manipulated randomly a submodular optimization instance, and measured how the output distribution of a differentially private soft-max (Power and Exponential mechanism with a given parameter) is affected by the manipulation (x-axis). In the y-axis we report the average objective obtained by the algorithm and parameter setting. The results in Figure 1 show that, for the same level of empirical sensitivity, the power mechanisms allows substantially improved results.

## 6.3 Sparse Multi-class Classification

Sparsity, or in our language worst-case approximation guarantee, is relevant both in multiclass classification and in designing attention mechanisms [14, 12]. As illustrated in Theorem 4.3, PLSOFT-MAX has small $\ell_q \to \ell_p$ smoothness for any $p, q$. In contrast, the mechanisms proposed in [14, 12] achieve much worse $\ell_q \to \ell_1$ smoothness as we can see below.

**Lemma 6.7.** *Let $h(\cdot) = \text{sparsegen-lin}(\cdot)$ be the generalization of $\text{sparsemax}(\cdot)$ function, then there exist $\boldsymbol{x}, \boldsymbol{y} \in \mathbb{R}^d$ such that $\|h(\boldsymbol{x}) - h(\boldsymbol{y})\|_1 \geq \frac{1}{2} d^{1-1/q} \|\boldsymbol{x} - \boldsymbol{y}\|_q$.*

In contrast, PLSOFTMAX achieves $\ell_q \to \ell_1$ smoothness of order $\min\{q + 1, \log(d)\}$. Smoothness is preferred for gradient calculation in commonly adopted stochastic gradient descent algorithms. To illustrate this we define a loss function with properties that are summarized in the following proposition. A detailed explanation of the loss function and a proof of Proposition 6.8 are presented in the the full version of the paper.

**Proposition 6.8.** *The exists a loss function $L_{\text{PLSOFTMAX}} : \mathbb{R}^d \times \Delta_{d-1} \to \mathbb{R}_+$ such that: (1) $L_{\text{PLSOFTMAX}}(\boldsymbol{x}; \boldsymbol{q}) \geq 0$, (2) $L_{\text{PLSOFTMAX}}(\boldsymbol{x}; \boldsymbol{q}) = 0 \Leftrightarrow \text{PLSOFTMAX}^\delta(\boldsymbol{x}) = \boldsymbol{q}$, (3) $L_{\text{PLSOFTMAX}}(\boldsymbol{x}; \boldsymbol{q})$ is a convex function with respect to $\boldsymbol{x}$.*

## Broader Impact

In this paper, we study some basic mathematical properties of soft-max functions and we propose new ones that are optimal with respect to some mathematical criteria. Soft-max functions are fundamental building blocks with many applications, from Machine Learning to Differential Privacy to Resource Allocation. All of these fields have societal impact: Differential Privacy has already been a fundamental mathematical tool to ensure privacy in the digital world and in many cases it has been the only available method to get privacy from services that take place in the digital world. Resource allocation and in particular auction theory has also societal impact, from the way that items are sold in online platforms to the way that ride-sharing applications decide prices, to the nation wide auctions that allocate bandwidth of the frequency spectrum to broadcast companies. Since our paper contributes to one of the fundamental tools in all these areas we believe that it can potentially have a positive impact by improving the outcomes of many algorithms in these topics in terms of e.g. privacy in Differential Privacy applications and revenue in Resource Allocation applications.

Although our paper is mostly mathematical in nature, we present also some experimental results applied to data collect from the DBLP dataset. Although we used a public data set, we acknowledge that the data we used may be biased.

## Funding Disclosure

The only funding to be declared is that MZ was supported by a Google Ph.D. Fellowship.

## Footnotes

[1] Note that throughout this section, we restrict the domain of the soft-max function to only positive values.

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
