[Supplementary Material]

# Supplementary Material for the Paper: Optimal Tradeoffs Between Utility and Smoothness for Soft-Max Functions

## A  Lower Bounds for the Exponential Mechanism

In this section, we prove Theorems 3.2 and 3.3.

*Proof of Theorem 3.2.* Fix a soft-max function $\boldsymbol{f} : \mathbb{R}^d \to \Delta_d$ that is $\delta$-approximate. It is well known that the Rényi Divergence of order $\alpha$ is a non-decreasing function of $\alpha$ for $\alpha \geq 1$. Hence it suffices to prove the statement of Theorem 3.2 for $\alpha = 1$ where $\mathrm{D}_\alpha$ become the KL-divergence $\mathrm{D}_{\mathrm{KL}}$. Observe also that without loss of generality we can assume that $\boldsymbol{f}$ is permutation invariant, i.e., for every permutation $\pi$ of $\{1, \ldots, d\}$ and every $x \in \mathbb{R}^d$, $f(\pi(x)) = \pi(f(x))$, where $\pi(x)$ denotes the vector $(x_{\pi(1)}, \ldots, x_{\pi(d)})$. If this is not the case then we can define the function $\boldsymbol{f}'$ which outputs the expectation of $\boldsymbol{f}$ over a random permutation of the coordinates of $\boldsymbol{x}$. It is easy to see then that $\boldsymbol{f}'$ has the same approximation and smoothness properties as $\boldsymbol{f}$ and is permutation invariant. Hence we assume that $\boldsymbol{f}$ is permutation invariant.

Let $a \in \mathbb{R}_+$. We define the vector $\boldsymbol{x}_a = (a, a, \ldots, a)^T$. For any $a$ because of the permutation invariance of $\boldsymbol{f}$ we have that $\boldsymbol{f}(\boldsymbol{x}_a) = (1/d, \ldots, 1/d)$. We define the vector $\boldsymbol{y}^{(a,b)}$ to be equal to $\boldsymbol{x}$ in all coordinates but 1 and equal to $b > a$ at the 1st coordinate. That is

$$y_j^{(a,b)} = a \quad \text{for } j \neq 1$$
$$\text{and} \quad y_1^{(a,b)} = b$$

From the approximation guarantee at $\boldsymbol{y}^{(a,b)}$ we have that

$$\left\| \boldsymbol{y}^{(a,b)} \right\|_\infty - \langle \boldsymbol{f}\left(\boldsymbol{y}^{(a,b)}\right), \boldsymbol{y}^{(a,b)} \rangle \leq \delta \implies$$
$$b - b f_1\left(\boldsymbol{y}^{(a,b)}\right) - a\left(1 - f_1\left(\boldsymbol{y}^{(a,b)}\right)\right) \leq \delta$$

Let $q = f_1\left(\boldsymbol{y}^{(a,b)}\right)$. Then we have

$$(b - a)(1 - q) \leq \delta.$$

This implies

$$q \geq 1 - \frac{\delta}{b - a}. \tag{A.1}$$

Also observe that because of the permutation invariance of $\boldsymbol{f}$ it holds that $f_i(\boldsymbol{y}^{(a,b)}) = (1-q)/(d-1)$ for any $i > 1$. Now we bound the KL-divergence of $\boldsymbol{f}$ when applied to the vectors $\boldsymbol{x}_a$ and $\boldsymbol{y}^{(a,b)}$:

$$
\begin{aligned}
\mathrm{D}_{\mathrm{KL}}\left(\boldsymbol{f}\left(\boldsymbol{y}^{(a,b)}\right) \| \boldsymbol{f}(\boldsymbol{x}_a)\right) &= \sum_{i=1}^{d} f_i(\boldsymbol{y}^{(a,b)}) \log\left(\frac{f_i(\boldsymbol{y}^{(a,b)})}{f_i(\boldsymbol{x}_a)}\right) \\
&= q \log(dq) + (1-q)\log\left((1-q)\frac{d}{d-1}\right) \\
&\geq q \log(d) - 1,
\end{aligned}
$$

where the last inequality follows from the fact that the binary entropy function $H(q) = -q\log(q) - (1-q)\log(1-q)$ is upper bounded by 1 and the fact that $\log(d) \geq \log(d-1)$. Using also A.1 we get that

$$
\mathrm{D}_{\mathrm{KL}}\left(\boldsymbol{f}\left(\boldsymbol{y}^{(a,b)}\right) \| \boldsymbol{f}(\boldsymbol{x}_a)\right) \geq \left(1 - \frac{\delta}{b-a}\right) \log(d) - 1.
$$

If we now set $b - a = 2\delta$ then we get $\left\|\boldsymbol{y}^{(a,b)} - \boldsymbol{x}_a\right\|_p = 2\delta$ and

$$
\mathrm{D}_{\mathrm{KL}}\left(\boldsymbol{f}\left(\boldsymbol{y}^{(a,b)}\right) \| \boldsymbol{f}(\boldsymbol{x}_a)\right) \geq \frac{1}{2} \log(d) - 1.
$$

Therefore,

$$
\frac{\mathrm{D}_{\mathrm{KL}}\left(\boldsymbol{f}\left(\boldsymbol{y}^{(a,b)}\right) \| \boldsymbol{f}(\boldsymbol{x}_a)\right)}{\left\|\boldsymbol{y}^{(a,b)} - \boldsymbol{x}_a\right\|_p} \geq \frac{\log(d) - 2}{4\delta}
$$

and the theorem follows. $\qquad\square$

*Proof of Theorem 3.3.* Let $\delta > 0$ and for the sake of contradiction assume that there exists a softmax function $\boldsymbol{f}$ that is both $\delta$-approximate in the worst-case and satisfies $(\ell_p, \mathrm{D}_\alpha)$-Lischitzness. We define $\boldsymbol{x} = (2\delta, 0, 0, \ldots, 0)$ and $\boldsymbol{y} = (0, 2\delta, 0, \ldots, 0)$ from the worst-case approximation guarantees of $\boldsymbol{f}$ we have that $\boldsymbol{f}(\boldsymbol{x}) = (1, 0, \ldots, 0)$, whereas $\boldsymbol{f}(\boldsymbol{y}) = (0, 1, 0, \ldots, 0)$. It is easy to see that for any $\alpha \geq 1$ it holds that $\mathrm{D}_\alpha\left(\boldsymbol{f}(\boldsymbol{x}) \| \boldsymbol{f}(\boldsymbol{x})\right) = \infty$ but $\left\|\boldsymbol{f}(\boldsymbol{x}) - \boldsymbol{f}(\boldsymbol{y})\right\|_p \leq 2$. The later contradicts the $(\ell_p, \mathrm{D}_\alpha)$-Lipschitzness of $\boldsymbol{f}$ and hence the theorem follows. $\qquad\square$

# B   The Construction of PLSOFTMAX

We first give an intuitive explanation of the proof of the construction. One notion that will be useful for this purpose in the following.

**Vector and Matrix Norms.** We define the $(\alpha, \beta)$-*subordinate norm* of a matrix $\boldsymbol{A} \in \mathbb{R}^{d \times \ell}$ to be

$$
\|\boldsymbol{A}\|_{\alpha,\beta} = \max_{\boldsymbol{x} \in \mathbb{R}^\ell, \boldsymbol{x} \neq 0} \|\boldsymbol{A}\boldsymbol{x}\|_\beta / \|\boldsymbol{x}\|_\alpha .
$$

The computation of $\|\boldsymbol{A}\|_{\alpha,\beta}$ is in general NP-hard and even hard to approximate, see [28, 16].

**Notation.** We use $\boldsymbol{E}_{i,j}$ to refer to the all zero matrix with one 1 at the $(i, j)$ entry.

The construction of PLSOFTMAX begins with the observation that for any $\boldsymbol{g} : \mathbb{R}^d \to \mathbb{R}^d$ and any $p, q \geq 1$, it holds that

$$
\|\boldsymbol{g}(\boldsymbol{x}) - \boldsymbol{g}(\boldsymbol{y})\|_q \leq \left(\max_{\boldsymbol{\xi} \in \mathbb{R}^d} \|\boldsymbol{J}_{\boldsymbol{g}}(\boldsymbol{\xi})\|_{p,q}\right) \|\boldsymbol{x} - \boldsymbol{y}\|_p
$$

where $\boldsymbol{J}_{\boldsymbol{g}}(\boldsymbol{\xi})$ is the Jacobian matrix of $\boldsymbol{g}$ at the point $\boldsymbol{\xi} \in \mathbb{R}^d$. Hence our goal is to construct a function $\boldsymbol{g}$ that does not violate the worst-case approximation conditions and for which we can also bound $\|\boldsymbol{J}_{\boldsymbol{g}}(\boldsymbol{\xi})\|_{p,q}$. To achieve this we carefully analyze the approximation conditions. Based on them we split the space $\mathbb{R}^d$ into small convex polytopes $P_i$ such that in each $P_i$, the approximation conditions do not change. Since, as we will see, the approximation condition is a linear condition, we choose our function $\boldsymbol{g}$ in $P_i$ to be a linear function that satisfies the approximation condition

44 inside the polytope $P_i$. Then we have to make sure that on the boundaries of $P_i$ the function is
45 continuous and that the $(p, q)$-subordinate norm of the matrices that we used in each $P_i$ is bounded
46 by some constant.

47 One important observation is that in each $P_i$, if some of the $[d]$ alternatives have low values, the
48 approximation constraint imposes that we cannot use at all any of these alternatives. Hence the
49 dimension of $P_i$ effectively becomes less than $d$. In these cases, we reduce the construction in $P_i$ to
50 a smaller dimensional construction that is solved inductively. We express this inductive argument as
51 a recursive relation over the matrices that is stated in Lemma B.4. Finally, one important theorem
52 that enables us to prove a precise bound on $\|\boldsymbol{J}_{\boldsymbol{g}}(\boldsymbol{\xi})\|_{p,1}$ is Theorem B.6. This is a generalization of
53 Theorem 1 of [10] which might be of independent interest.

54 Now that we described the high level idea of our construction, we dive in to the technical details.
55 The function $\boldsymbol{f}$ that we are going to construct is a piece-wise linear function. So we first define the
56 notion of a piece-wise linear function in $d$ dimensions.

57 **Definition B.1** (PIECE-WISE LINEAR FUNCTIONS). *A function $\boldsymbol{f} : \mathbb{R}^d \to \mathbb{R}^d$ is piece-wise linear*
58 *if there exist a finite partition $\mathcal{P}_{\boldsymbol{f}} = \{P_1, \ldots, P_L\}$ of $\mathbb{R}^d$ such that $P_i$ is a convex polytope, for any*
59 *$i$ and any $\boldsymbol{x} \in P_i$ there exists a unique matrix $\boldsymbol{A}_i \in \mathbb{R}^{d \times d}$ and a unique vector $\boldsymbol{b}_i \in \mathbb{R}^d$ such that*

$$\boldsymbol{f}(\boldsymbol{x}) = \boldsymbol{A}_i \boldsymbol{x} + \boldsymbol{b}_i.$$

60 *We use $\mathcal{A}_{\boldsymbol{f}}$ to refer to the set of matrices $\{\boldsymbol{A}_1, \ldots, \boldsymbol{A}_L\}$.*

61 Our construction proceeds in the following steps:

62      1. define the partition $\mathcal{P}_{\boldsymbol{f}}$ of $\mathbb{R}^d$, the matrix $\boldsymbol{A}_i$, and vector $\boldsymbol{b}_i$ that we use for every $P_i \in \mathcal{P}_{\boldsymbol{f}}$,

63      2. describe the set $\mathcal{A}_{\boldsymbol{f}}$ and its properties,

64      3. prove that the defined $\boldsymbol{f}$ is continuous on the boundaries of $P_i$'s,

65      4. prove that it has small absolute approximation loss, and

66      5. prove that $\|\boldsymbol{A}_i\|_{p,1}$ is small and hence using Lemma B.2 conclude that $\boldsymbol{f}$ is has small
67         Lipschitz constant.

68 For simplicity of the proof we will use $\boldsymbol{f}$ to refer to PLSOFTMAX$^\delta$ within the scope of this section.

## B.1 Piece-wise linear functions

70 For piece-wise linear functions $\boldsymbol{f}$, we use the following lemma to establish the Lipschitz property.

71 **Lemma B.2.** *Let $\boldsymbol{f} : \mathbb{R}^d \to \mathbb{R}^d$ be a continuous and piece-wise linear function and let $p, q \geq 1$,*
72 *then*

$$\|\boldsymbol{f}(\boldsymbol{x}) - \boldsymbol{f}(\boldsymbol{y})\|_q \leq \left( \max_{\boldsymbol{A} \in \mathcal{A}_{\boldsymbol{f}}} \|\boldsymbol{A}\|_{p,q} \right) \cdot \|\boldsymbol{x} - \boldsymbol{y}\|_p \quad \forall \boldsymbol{x}, \boldsymbol{y} \in \mathbb{R}^d$$

73 *Proof.* We first prove the single variable case, that is, we prove that for any continuous piece-wise
74 linear function $\boldsymbol{g} : \mathbb{R} \to \mathbb{R}^d$ and if $c = \max_{\boldsymbol{A} \in \mathcal{A}_{\boldsymbol{g}}} \|\boldsymbol{A}\|_{p,q}$ then for any $x, y \in \mathbb{R}$

$$\|\boldsymbol{g}(x) - \boldsymbol{g}(y)\|_q \leq c |x - y|.$$

75 Without loss of generality assume that $x > y$. Since $\boldsymbol{g}$ is piece-wise linear, we have a sequence $y =$
76 $x_1 < x_2 < \cdots < x_L = x$ such that for any $z \in [x_i, x_{i+1}] : \boldsymbol{g}(z) = \boldsymbol{a}_i z + \boldsymbol{b}_i$ for some $\boldsymbol{a}_i, \boldsymbol{b}_i \in \mathbb{R}^d$.
77 Also notice that since $\boldsymbol{a}_i$ is a vector, by definition of subordinate norms, $\|\boldsymbol{a}_i\|_{p,q} = \|\boldsymbol{a}_i\|_q$. Now
78 because of the continuity of $\boldsymbol{g}$

$$\|\boldsymbol{g}(x) - \boldsymbol{g}(y)\|_q \leq \sum_{i=1}^{L-1} \|\boldsymbol{g}(x_{i+1}) - \boldsymbol{g}(x_i)\|_q = \sum_{i=1}^{L-1} \|\boldsymbol{a}_i(x_{i+1} - x_i)\|_q = \sum_{i=1}^{L-1} \|\boldsymbol{a}_i\|_q (x_{i+1} - x_i)$$

$$\leq c \left( \sum_{i=1}^{L-1} (x_{i+1} - x_i) \right) = c(x - y).$$

79   For the general case, let $c = \max_{\boldsymbol{A} \in \mathcal{A}_f} \|\boldsymbol{A}\|_{p,q}$ and $\boldsymbol{x}, \boldsymbol{y} \in \mathbb{R}^d$. We define the following function

80   $\boldsymbol{h} : [0,1] \to \mathbb{R}^d$ which is easy to verify that is also continuous and piece-wise linear:

$$\boldsymbol{h}(t) = \boldsymbol{f}\left(t\boldsymbol{x} + (1-t)\boldsymbol{y}\right).$$

There exists a sequence $0 = t_1 < t_2 < \cdots < t_L = 1$, such that for every $i$, the function $f$ has a linear form $f(\boldsymbol{u}) = A_i \boldsymbol{u} + b_i$ on the set $\{t\boldsymbol{x} + (1-t)\boldsymbol{y} : \ t \in [t_i, t_{i+1}]\}$. Therefore, for every $t \in [t_i, t_{i+1}]$, by the definition of $\boldsymbol{h}$,

$$\boldsymbol{h}(t) = \boldsymbol{A}_i(t\boldsymbol{x} + (1-t)\boldsymbol{y}) + \boldsymbol{b}_i = \boldsymbol{A}_i(\boldsymbol{x} - \boldsymbol{y})t + \boldsymbol{b}_i + \boldsymbol{A}_i\boldsymbol{y}.$$

81   Therefore, on $t \in [t_i, t_{i+1}]$, the function $\boldsymbol{h}$ has the linear form $\boldsymbol{h}(t) = \boldsymbol{v}_i t + \boldsymbol{w}_i$ for $\boldsymbol{v}_i = \boldsymbol{A}_i(\boldsymbol{x} - \boldsymbol{y})$

82   and $\boldsymbol{w}_i = \boldsymbol{A}_i\boldsymbol{y} + \boldsymbol{b}_i$. Hence by the definition of the subordinate matrix norm we have that

$$\|\boldsymbol{v}_i\|_q = \|\boldsymbol{A}_i(\boldsymbol{x} - \boldsymbol{y})\|_q \leq \|\boldsymbol{A}_i\|_{p,q} \|\boldsymbol{x} - \boldsymbol{y}\|_p \leq c \|\boldsymbol{x} - \boldsymbol{y}\|_p.$$

83   Since $i$ was arbitrary we have that $c' = \max_{\boldsymbol{A} \in \mathcal{A}_h} \|\boldsymbol{A}\|_{p,q} \leq c \|\boldsymbol{x} - \boldsymbol{y}\|_p$. Finally using the state-

84   ment of the lemma for the single variable case that we already proved, we have that

$$\|\boldsymbol{f}(\boldsymbol{x}) - \boldsymbol{f}(\boldsymbol{y})\|_q = \|\boldsymbol{h}(1) - \boldsymbol{h}(0)\|_q \leq c'(1-0) \leq c \|\boldsymbol{x} - \boldsymbol{y}\|_p.$$

85                                                               □

## B.2   Properties of the Soft-Max Matrices

87   Recall the definition of the soft max matrices in Section 4.

88   **Definition B.3** (SOFT-MAX MATRICES). The soft max matrix $\boldsymbol{SM}_{(k,d)} = (m_{ij}) \in \mathbb{R}^{d \times d}$ with

89   parameters $k$, $d$ is defined as follows

$$m_{11} = \frac{k-1}{k} \tag{B.1}$$

$$m_{ii} = \frac{1}{i} \qquad\qquad\qquad \forall i \in [2,k] \tag{B.2}$$

$$m_{i1} = -\frac{1}{k} \qquad\qquad\qquad \forall i \in [2,k] \tag{B.3}$$

$$m_{ij} = \frac{1}{j} - \frac{1}{j-1} \qquad\qquad\qquad \forall j > i, j \in [2,k] \tag{B.4}$$

$$m_{ij} = 0 \qquad\qquad \forall i,j \text{ s.t. } (i \in [k+1,d]) \vee (j \in [k+1,d]) \tag{B.5}$$

90   Schematically we have

$$\boldsymbol{SM}_{(k,d)} = \begin{pmatrix}
\frac{k-1}{k} & -\frac{1}{2} & -\frac{1}{6} & \cdots & -\frac{1}{k(k-1)} & 0 & \cdots & 0 \\
-\frac{1}{k} & \frac{1}{2} & -\frac{1}{6} & \cdots & -\frac{1}{k(k-1)} & 0 & \cdots & 0 \\
-\frac{1}{k} & 0 & \frac{1}{3} & \cdots & -\frac{1}{k(k-1)} & 0 & \cdots & 0 \\
-\frac{1}{k} & 0 & 0 & \cdots & -\frac{1}{k(k-1)} & 0 & \cdots & 0 \\
\vdots & \vdots & \vdots & \ddots & \vdots & \vdots & \ddots & \vdots \\
-\frac{1}{k} & 0 & 0 & \cdots & \frac{1}{k} & 0 & \cdots & 0 \\
0 & 0 & 0 & \cdots & 0 & 0 & \cdots & 0 \\
\vdots & \vdots & \vdots & \ddots & \vdots & \vdots & \ddots & \vdots \\
0 & 0 & 0 & \cdots & 0 & 0 & \cdots & 0
\end{pmatrix}$$

91   We also define the columns and the rows of the soft max matrices as follows

$$\boldsymbol{SM}_{(k,d)} = \begin{pmatrix} | & | & & | & | & & | \\ \boldsymbol{m}_1^{(k,d)} & \boldsymbol{m}_2^{(k,d)} & \cdots & \boldsymbol{m}_k^{(k,d)} & \boldsymbol{0} & \cdots & \boldsymbol{0} \\ | & | & & | & | & & | \end{pmatrix} \tag{B.6}$$

$$
\boldsymbol{SM}_{(k,d)} = \begin{pmatrix}
- & \left(\boldsymbol{s}_1^{(k,d)}\right)^T & - \\
- & \left(\boldsymbol{s}_2^{(k,d)}\right)^T & - \\
 & \vdots & \\
- & \left(\boldsymbol{s}_k^{(k,d)}\right)^T & - \\
- & \boldsymbol{0} & - \\
 & \vdots & \\
- & \boldsymbol{0} & -
\end{pmatrix}
\tag{B.7}
$$

Below are some examples for $d = 4$.

$$
\boldsymbol{SM}_{(1,4)} = \begin{pmatrix}
0 & 0 & 0 & 0 \\
0 & 0 & 0 & 0 \\
0 & 0 & 0 & 0 \\
0 & 0 & 0 & 0
\end{pmatrix}
\qquad
\boldsymbol{SM}_{(2,4)} = \begin{pmatrix}
\frac{1}{2} & -\frac{1}{2} & 0 & 0 \\
-\frac{1}{2} & \frac{1}{2} & 0 & 0 \\
0 & 0 & 0 & 0 \\
0 & 0 & 0 & 0
\end{pmatrix}
$$

$$
\boldsymbol{SM}_{(3,4)} = \begin{pmatrix}
\frac{2}{3} & -\frac{1}{2} & -\frac{1}{6} & 0 \\
-\frac{1}{3} & \frac{1}{2} & -\frac{1}{6} & 0 \\
-\frac{1}{3} & 0 & \frac{1}{3} & 0 \\
0 & 0 & 0 & 0
\end{pmatrix}
\qquad
\boldsymbol{SM}_{(4,4)} = \begin{pmatrix}
\frac{3}{4} & -\frac{1}{2} & -\frac{1}{6} & -\frac{1}{12} \\
-\frac{1}{4} & \frac{1}{2} & -\frac{1}{6} & -\frac{1}{12} \\
-\frac{1}{4} & 0 & \frac{1}{3} & -\frac{1}{12} \\
-\frac{1}{4} & 0 & 0 & \frac{1}{4}
\end{pmatrix}
$$

Now we prove some properties of the soft max matrices, that will help us latex prove the continuity and the smoothness of PLSOFTMAX.

**Lemma B.4.** *For any $d \in \mathbb{N}$ and $k \in [d]$ the following recursive relation holds*
$$
\boldsymbol{SM}_{(k-1,d)} = \boldsymbol{SM}_{(k,d)}\left(\boldsymbol{I} + \boldsymbol{E}_{k,1} - \boldsymbol{E}_{k,k}\right)
$$

*Proof.* From (B.6) we have that
$$
\boldsymbol{SM}_{(k,d)}\left(\boldsymbol{I} + \boldsymbol{E}_{k,1} - \boldsymbol{E}_{k,k}\right) = \begin{pmatrix}
\boldsymbol{m}_1^{(k,d)} + \boldsymbol{m}_k^{(k,d)} & \boldsymbol{m}_2^{(k,d)} & \cdots & \boldsymbol{m}_{k-1}^{(k,d)} & \boldsymbol{0} & \cdots & \boldsymbol{0}
\end{pmatrix}.
$$

We now observe by the definition of the soft max matrices that for any $d \in \mathbb{N}$, $k, k' \in [d]$ and $j \in [2, \min\{k, k'\}]$ it holds that $\boldsymbol{m}_j^{(k,d)} = \boldsymbol{m}_j^{(k',d)}$. Hence we only have to prove that
$$
\boldsymbol{m}_1^{(k-1,d)} = \boldsymbol{m}_1^{(k,d)} + \boldsymbol{m}_k^{(k,d)}
$$
and the lemma follows. For this we have that
$$
m_{11}^{(k,d)} + m_{1k}^{(k,d)} = \frac{k-1}{k} - \frac{1}{k(k-1)} = \frac{(k-1)^2 - 1}{k(k-1)} = \frac{k-2}{k-1} = m_{11}^{(k-1,d)}
$$
also for $i \in [2, k-1]$ we have that
$$
m_{i1}^{(k,d)} + m_{ik}^{(k,d)} = -\frac{1}{k} - \frac{1}{k(k-1)} = -\frac{1}{k-1} = m_{i1}^{(k-1,d)}
$$
and finally
$$
m_{k1}^{(k,d)} + m_{kk}^{(k,d)} = -\frac{1}{k} + \frac{1}{k} = 0 = m_{k1}^{(k-1,d)}
$$
and the lemma follows. $\qquad\square$

**Lemma B.5.** *Let $r, t \in [d]$ with $r > t$ and $\boldsymbol{x} \in \mathbb{R}^d$ be a vector with the property that $x_i = x_j = x$ for any $i, j \in [r, t]$ then the vector $\boldsymbol{y} \in \mathbb{R}^d$ with*
$$
\boldsymbol{y} = \boldsymbol{SM}_{(k,d)}\boldsymbol{x}
$$
*has also the property $y_i = y_j$ for any $i, j \in [r, t]$.*

*Proof.* From (B.7) we have that

$$y = SM_{(k,d)}x = \begin{pmatrix} s_1^T x \\ s_2^T x \\ \vdots \\ s_k^T x \\ 0 \\ \vdots \\ 0 \end{pmatrix}$$

where for simplicity we dropped the indicators $(k, d)$ from the row vectors $s_i$ since we keep $k, d$ constant through the proof. Therefore we have that

$$\begin{pmatrix} y_r \\ y_{r+1} \\ \vdots \\ y_t \end{pmatrix} = \begin{pmatrix} \sum_{j=1}^{r-1} s_{rj}x_j + \left(\sum_{j=r}^{t} s_{rj}\right)x + \sum_{j=t+1}^{d} s_{rj}x_j \\ \sum_{j=1}^{r-1} s_{(r+1)j}x_j + \left(\sum_{j=r}^{t} s_{(r+1)j}\right)x + \sum_{j=t+1}^{d} s_{(r+1)j}x_j \\ \vdots \\ \sum_{j=1}^{r-1} s_{tj}x_j + \left(\sum_{j=r}^{t} s_{tj}\right)x + \sum_{j=t+1}^{d} s_{tj}x_j \end{pmatrix}$$

but by the definition of the soft max matrices we can easily see that for any $i, i' \in [r, t]$ and $j < r$ or $j > t$ it holds that $s_{ij} = s_{i'j}$. This observation together with the above calculations imply that it suffices to prove that for any $i, i' \in [r, t]$ it holds that

$$\sum_{j=r}^{t} s_{ij} = \sum_{j=r}^{t} s_{i'j} \tag{B.8}$$

also because of the symmetry of the zero entries of soft max matrices for $t > k$ it suffices to prove this statement for $t \leq k$. We also consider two case $r = 1$ and $r > 1$.

**$r = 1$.** For $i = 1$ we have that

$$\sum_{j=r}^{t} s_{1j} = s_{11} + \sum_{j=2}^{t} s_{1j} = \frac{k-1}{k} - \sum_{j=2}^{t} \frac{1}{j(j-1)}$$

and using the following relation

$$\sum_{j=n}^{m} \frac{1}{j(j-1)} = \sum_{j=n}^{m} \left(\frac{1}{j-1} - \frac{1}{j}\right) = \frac{1}{m-1} - \frac{1}{n} \tag{B.9}$$

we get that

$$\sum_{j=r}^{t} s_{1j} = \frac{k-1}{k} - \left(1 - \frac{1}{t}\right) = \frac{1}{t} - \frac{1}{k}.$$

For $i > 1$ we have that

$$\sum_{j=r}^{t} s_{ij} = s_{i1} + s_{ii} + \sum_{j=i+1}^{t} s_{ij} = -\frac{1}{k} + \frac{1}{i} - \sum_{j=i+1}^{t} \frac{1}{j(j-1)} \overset{(B.9)}{=} -\frac{1}{k} + \frac{1}{i} - \left(\frac{1}{i} - \frac{1}{t}\right) = \frac{1}{t} - \frac{1}{k}.$$

Hence the sum $\sum_{j=1}^{t} s_{ij}$ does not depend on $i$ and the property (B.8) holds for $r = 1$.

**$r > 1$.** For any $i \in [r, t]$ we have that

$$\sum_{j=r}^{t} s_{ij} = s_{ii} + \sum_{j=i+1}^{t} s_{ij} = \frac{1}{i} - \sum_{j=i+1}^{t} \frac{1}{j(j-1)} \overset{(B.9)}{=} \frac{1}{i} - \left(\frac{1}{i} - \frac{1}{t}\right) = \frac{1}{t}$$

and again we observe that the sum $\sum_{j=r}^{t} s_{ij}$ does not depend on $i$ and the property (B.8) holds for any $r > 1$, $r \leq t$. This implies $y_r = \cdots = y_t$ and the lemma follows. $\qquad\square$

122 Finally our goal is to bound $\left\|\boldsymbol{SM}_{(k,d)}\right\|_{p,q}$ for any $p, q \in [1, \infty]$. Before that we give a proof of a
123 general property of the subordinate norm $\|\cdot\|_{p,1}$. This corresponds to the following generalization of
124 Theorem 1 in [10]. Drakakis and Pearlmutter [10] only state the result for the $\|\cdot\|_{2,1}$ norm although
125 their proof generalizes.

126 **Theorem B.6** (Generalization of Theorem 1 [10])**.** *Let* $\boldsymbol{A} \in \mathbb{R}^{t \times d}$ *and* $p \in 2\mathbb{N}_+$*, then*

$$\|\boldsymbol{A}\|_{p,1} = \max_{\boldsymbol{s} \in \{-1,1\}^t} \left\|\boldsymbol{s}^T \boldsymbol{A}\right\|_r \quad where \ r = \frac{p}{p-1}.$$

127 *In particular the* $\ell_r$ *norm is the dual norm of the* $\ell_p$ *norm.*

128 *Proof of Theorem B.6.* Let $\boldsymbol{a}_i^T$ be the $i$ th row of the matrix $\boldsymbol{A}$. By the definition of the subordinate
129 norm we have that

$$\|\boldsymbol{A}\|_{p,1} = \max_{\boldsymbol{x} \in \mathbb{R}^d, \|\boldsymbol{x}\|_p = 1} \|\boldsymbol{A}\boldsymbol{x}\|_1 .$$

130 We first prove that the maximum of the above optimization problem lies in a region of the space
131 where $\boldsymbol{a}_i^T \boldsymbol{x} \neq 0$ for all $i \in [t]$. This implies that we can find the maximum in a subspace of the
132 space where both the objective and the constraint are differentiable and hence we can use first order
133 conditions to determine the maximum. This is described in the following claim.

134 **Claim B.7.** *Let*

$$\boldsymbol{x} = \arg \max_{\boldsymbol{y} \in \mathbb{R}^d, \|\boldsymbol{y}\|_p = 1} \|\boldsymbol{A}\boldsymbol{y}\|_1$$

135 *then for every* $i \in [t]$ *holds that* $\boldsymbol{a}_i^T \boldsymbol{x} \neq 0$.

136 *Proof.* We prove this claim by contradiction. Let's assume without loss of generality that for $i =$
137 $1, \ldots, \ell$ its true that $\boldsymbol{a}_i^T \boldsymbol{x} = 0$, where $\ell \in [t]$. Then we define the vector $\boldsymbol{z}$ as

$$\boldsymbol{z} = \frac{\boldsymbol{x} + \eta \boldsymbol{a}_1}{\|\boldsymbol{x} + \eta \boldsymbol{a}_1\|_p}$$

138 with $\eta$ that can be either positive or negative and is small enough so that $\operatorname{sign}(\boldsymbol{a}_i^T \boldsymbol{x}) = \operatorname{sign}(\boldsymbol{a}_i^T \boldsymbol{z})$.
139 We define the following real valued function $h : \mathbb{R} \to \mathbb{R}$ as $h(\eta) = 1/\|\boldsymbol{x} + \eta \boldsymbol{a}_1\|_p$. It is easy to
140 see that the first and the second derivative of $h$ for $\eta$ in the interval $[-1, 1]$ are bounded. Hence by
141 Taylor's theorem we have that

$$h(\eta) = h(0) + h'(0)\eta + O(\eta^2).$$

142 By simple calculations it is also easy to see that $h(0) = 1$ and $h'(0) = \sum_{i=1}^{t} a_{1i} x_i^{p-1}$. Let also
143 $s_i = \operatorname{sign}(\boldsymbol{a}_i^T \boldsymbol{x})$. This implies

$$\sum_{i=1}^{t} \left|\boldsymbol{a}_i^T \boldsymbol{z}\right| = \left(\sum_{i=1}^{t} \left|\boldsymbol{a}_i^T \boldsymbol{x}\right| + |\eta| \sum_{i=1}^{\ell} \left|\boldsymbol{a}_i^T \boldsymbol{a}_1\right| + \eta \sum_{i=\ell+1}^{t} s_i \boldsymbol{a}_i^T \boldsymbol{a}_1\right) (h(0) + h'(0)\eta + O(\eta^2))$$

$$= \sum_{i=1}^{t} \left|\boldsymbol{a}_i^T \boldsymbol{x}\right| + |\eta| \sum_{i=1}^{\ell} \left|\boldsymbol{a}_i^T \boldsymbol{a}_1\right| + \left(\sum_{i=\ell+1}^{t} s_i \boldsymbol{a}_i^T \boldsymbol{a}_1 + \sum_{i=1}^{t} \left|\boldsymbol{a}_i^T \boldsymbol{x}\right|\right) \eta + O(\eta^2)$$

$$= \sum_{i=1}^{t} \left|\boldsymbol{a}_i^T \boldsymbol{x}\right| + C_1 |\eta| + C_2 \eta + O(\eta^2)$$

144 Now without loss of generality we can assume that $\boldsymbol{a}_1 \neq \boldsymbol{0}$ and hence $C_1 > 0$. Also choosing the
145 correct sign for $\eta$ we can have $C_2 \eta \geq 0$. Finally we can make $\eta$ small enough so that $C_1 |\eta| + C_2 \eta +$
146 $O(\eta^2) > 0$ and hence $\sum_{i=1}^{t} \left|\boldsymbol{a}_i^T \boldsymbol{z}\right| > \sum_{i=1}^{t} \left|\boldsymbol{a}_i^T \boldsymbol{x}\right|$ which contradicts the assumption that $\boldsymbol{x}$ was the
147 maximum and the claim follows. $\qquad\square$

148 Using Claim B.7 we can see that the maximum of the program $\left(\max_{\boldsymbol{x} \in \mathbb{R}^d, \|\boldsymbol{x}\|_p = 1} \|\boldsymbol{A}\boldsymbol{x}\|_1\right)$ is
149 achieved for a vector that belongs to an open subset of the space where both the constraint and

the objective function are differentiable. Notice that the differentiability of the constraint follows from the fact that $p$ is an even number.

Using Langragian multipliers we can find the solution to this optimization problem using first order conditions on the following function

$$g(\boldsymbol{x}, \lambda) = \sum_{i=1}^{t} \left| \sum_{j=1}^{d} a_{ij} x_j \right| + \lambda \left( \|x\|_p - 1 \right)$$

which using the definition $s_i = \text{sign}(\boldsymbol{a}_i^T \boldsymbol{x})$ takes the form

$$g(\boldsymbol{x}, \lambda) = \sum_{i=1}^{t} s_i \sum_{j=1}^{d} a_{ij} x_j + \lambda \left( \|x\|_p - 1 \right).$$

We now compute the partial derivative of $g$ with respect to $x_k$ for some $k \in [d]$.

$$\frac{\partial g}{\partial x_k} = \sum_{i=1}^{t} s_i a_{ik} + \lambda \frac{x_k^{p-1}}{\|\boldsymbol{x}\|_p^{p-1}} = \sum_{i=1}^{t} s_i a_{ik} + \lambda x_k^{p-1}$$

hence $\frac{\partial g}{\partial x_k} = 0$ implies

$$x_k = -\frac{1}{\lambda^{1/(p-1)}} \left( \sum_{i=1}^{t} s_i a_{ik} \right)^{1/(p-1)} \tag{B.10}$$

and therefore

$$\|\boldsymbol{x}\|_p = \frac{1}{|\lambda|^{1/(p-1)}} \left\| \boldsymbol{s}^T \boldsymbol{A} \right\|_{p/(p-1)}^{1/(p-1)}.$$

From the constraint $\frac{\partial g}{\partial \lambda} = 0$ we get that

$$|\lambda| = \left\| \boldsymbol{s}^T \boldsymbol{A} \right\|_{p/(p-1)}.$$

Using (B.10) and the definition of the function $g$ we have that

$$g(\boldsymbol{x}, \lambda) = \sum_{i=1}^{t} s_i \sum_{j=1}^{d} a_{ij} x_j = \sum_{j=1}^{d} \left( \sum_{i=1}^{t} s_i a_{ij} \right) x_j$$

$$\overset{(B.10)}{=} \sum_{j=1}^{d} \left( -\lambda x_j^{p-1} \right) x_j$$

$$= -\lambda \sum_{j=1}^{d} x_j^p = \left\| \boldsymbol{s}^T \boldsymbol{A} \right\|_r$$

where $r = \frac{p}{p-1}$, and the theorem follows. $\qquad\square$

**Lemma B.8.** *For any $d \in \mathbb{N}$, $k \in [d]$ and $p, q \in [1, \infty]$ we have that*

$$\left\| \boldsymbol{SM}_{(k,d)} \right\|_{p,q} \leq 2 \min \left\{ p+1, \frac{q}{q-1}, \log(k) \right\}.$$

*Proof.* It is easy to see from the definition that $\left\| \boldsymbol{SM}_{(k,d)} \right\|_{p,q} = \left\| \boldsymbol{SM}_{(k,k)} \right\|_{p,q}$. Hence we can restrict our attention to the matrices $\boldsymbol{SM}_{(k,k)}$ which for simplicity we call $\boldsymbol{SM}_k$.

Our first goal is to prove for even $p$ that $\|\boldsymbol{SM}_k\|_{p,1} \leq 2p$ and since $\|\boldsymbol{x}\|_{p-1} \geq \|\boldsymbol{x}\|_p$ we can conclude that $\|\boldsymbol{SM}_k\|_{p-1,1} \leq \|\boldsymbol{SM}_k\|_{p,1} \leq 2p$. This implies $\|\boldsymbol{SM}_k\|_{p,1} \leq 2(p+1)$ for any $p$.

**Claim B.9.** *It holds that $\|\boldsymbol{SM}_k\|_{p,1} \leq 2(p+1)$ for any $p \in [1, \infty]$.*

*Proof.* Using the Theorem B.6 and setting $r = p/(p-1)$ we have that

$$\left\| \boldsymbol{SM}_k \right\|_{p,1} = \max_{\boldsymbol{z} \in \{-1,1\}^k} \left\| \boldsymbol{z}^T \boldsymbol{SM}_k \right\|_p.$$

Now for every column $\boldsymbol{m}_i$ of $\boldsymbol{SM}_k$ we observe that the sum of the coordinates is zero, that is $\sum_{j=1}^k m_{ji} = 0$. Also all the element except the diagonal elements are non-positive and hence it is true that

$$\sum_{j=1}^k |m_{ji}| = 2m_{ii}.$$

But obviously $\left| \boldsymbol{z}^T \boldsymbol{m}_i \right| \leq \sum_{j=1}^k |m_{ji}|$ for all $\boldsymbol{z} \in \{-1,1\}^k$. This implies that $\left| \boldsymbol{z}^T \boldsymbol{m}_i \right| \leq 2m_{ii} = 2/i$. Therefore for any $\boldsymbol{z} \in \{-1,1\}^k$ we have that

$$\left\| \boldsymbol{z}^T \boldsymbol{SM}_k \right\|_r = \left( \sum_{i=1}^k \left| \boldsymbol{z}^T \boldsymbol{m}_i \right|^r \right)^{1/r} \leq 2 \left( \sum_{i=1}^k \frac{1}{i^r} \right)^{1/r} \leq 2 \left( \zeta(r) \right)^{1/r} \tag{B.11}$$

where $\zeta(x)$ is the Riemann zeta function evaluated at $x$. Now we use the formula (2.1.16) of Chapter 2.1 of [31] and we get that

$$\zeta \left( \frac{p}{p-1} \right) \leq p.$$

This implies that

$$\left\| \boldsymbol{z}^T \boldsymbol{SM}_k \right\|_r \leq 2p^{(p-1)/p} \leq 2p.$$

This holds for any even $p$ since only in this case we can use Theorem B.6, and this implies that for any $p$

$$\left\| \boldsymbol{z}^T \boldsymbol{SM}_k \right\|_r \leq 2p^{(p-1)/p} \leq 2(p+1)$$

as we argued in the beginning of the proof. $\qquad\square$

Now it is obvious that $\|\cdot\|_{p,q} \leq \|\cdot\|_{p,1}$ and hence we have that $\left\| \boldsymbol{SM}_{(k,d)} \right\|_{p,q} \leq 2(p+1)$.

Also, for any $p$ and any vector $\boldsymbol{v} \in \mathbb{R}^d$, we have $\max_{\boldsymbol{x}:\|x\|_p=1} \left| \boldsymbol{v}^T \boldsymbol{x} \right| = \|\boldsymbol{v}\|_{q/(q-1)}$. Applying this on rows of any matrix $A$, we get

$$\|A\|_{p,q} \leq \left( \sum_i \|\boldsymbol{a}_i\|_{p/(p-1)}^q \right)^{1/q}.$$

Therefore, for every $q > 1$, and using the formula (2.1.16) of Chapter 2.1 of [31] and we get that

$$\left\| \boldsymbol{SM}_{(k,d)} \right\|_{p,q} \leq \left( \sum_{i=1}^k \frac{1}{i^q} \right)^{1/q} < \zeta(q)^{1/q} < \frac{q}{q-1}.$$

Finally we can use (B.11) and see that for any $q, p$

$$\left\| \boldsymbol{z}^T \boldsymbol{SM}_k \right\|_q \leq 2 \left( \sum_{i=1}^k \frac{1}{i} \right) \leq 2 \log k$$

and this completes the proof of the lemma. $\qquad\square$

## B.3 Proof of Theorem 4.3

We first prove that $\boldsymbol{f}$ is continuous and that its output is always a probability distribution over the $d$ coordinates, i.e. that its output belongs to $\Delta_{d-1}$.

**Continuity of $\boldsymbol{f}$.** From the definition of $\boldsymbol{f}$ is easy to see that $\boldsymbol{f}$ is piecewise linear, since it remains linear for all the regions where the order of the coordinates of $\boldsymbol{x}$ is fixed and $k_{\boldsymbol{x}}$ is fixed. It is easy to see that the set of these regions is a finite set and each region is a convex set. More formaly

$$\mathcal{P}_{\boldsymbol{f}} = \left\{ \left\{ \boldsymbol{x} \mid \left( x_{\pi(1)} \geq x_{\pi(2)} \geq \cdots \geq x_{\pi(d)} \right) \wedge \left( x_{\pi(1)} - x_{\pi(k)} \leq \delta \right) \right\} \mid \pi : [d] \to [d], k \in [d] \right\}$$

191   where $\pi$ has to be a permutation. Also the set of matrices that $\boldsymbol{f}$ uses is the following

$$\mathcal{A}_{\boldsymbol{f}} = \left\{ \frac{1}{\delta} \boldsymbol{P}^T \boldsymbol{SM}_{(k,d)} \boldsymbol{P} \mid k \in \mathbb{N}, \boldsymbol{P} \text{ permutation matrix} \right\}.$$

192   So its is clear that $\boldsymbol{f}$ is piecewise linear, but it is not clear that it should be continuous. To prove the
193   continuity of $\boldsymbol{f}$ we will use the Lemmas B.4, B.5. Since $\boldsymbol{f}$ is piecewise linear the only regions where
194   $\boldsymbol{f}$ might not be continuous are the boundaries of the regions $P_i \in \mathcal{P}_{\boldsymbol{f}}$. There are two types of such
195   boundaries one because of the change of the value $k_{\boldsymbol{x}}$ and because the ordering in $\boldsymbol{x}$ changes. First
196   consider the boundaries because of the change of $k_{\boldsymbol{x}}$ which for simplicity we call $k$ for the proof.
197   At the boundaries where $k$ decreases we have that $x_1 - x_k = \delta$ which implies $x_k = x_1 - \delta$. If we
198   apply this in the definition of $\boldsymbol{f}$, then we get

$$\boldsymbol{f}(\boldsymbol{x}) = \frac{1}{\delta} \boldsymbol{SM}_{(k,d)} \cdot \boldsymbol{x} + \boldsymbol{u}_k = \frac{1}{\delta} \left( \boldsymbol{SM}_{(k,d)} \left( \boldsymbol{I}_d + \boldsymbol{E}_{k,1} - \boldsymbol{E}_{k,k} \right) \right) \boldsymbol{x} + \frac{1}{\delta} \delta \boldsymbol{m}_k^{(k,d)} + \boldsymbol{u}_k$$
$$= \frac{1}{\delta} \boldsymbol{SM}_{(k-1,d)} \cdot \boldsymbol{x} + \boldsymbol{m}_k^{(k,d)} + \boldsymbol{u}_k$$
$$= \frac{1}{\delta} \boldsymbol{SM}_{(k-1,d)} \cdot \boldsymbol{x} + \boldsymbol{u}_{k-1}$$

199   where at the second step we used Lemma B.4. This implies that at these boundaries the function
200   remains continuous. The transition for $k$ to higher $k$ can be proved exactly the same way. Now we
201   consider the case where the ordering of $\boldsymbol{x}$ changes. In this case we will have that for any two indices
202   $i, j \in [d]$ that are changing order it is true that $x_i = x_j$. But from B.5 and the definition of $\boldsymbol{f}(\boldsymbol{x})$ we
203   have that $f_i(\boldsymbol{x}) = f_j(\boldsymbol{x})$. This implies that the relative order of $x_i$ and $x_j$ does not change the value
204   of $\boldsymbol{f}$. Hence in the boundaries where the coordinates of $\boldsymbol{x}$ change order $\boldsymbol{f}$ is continuous. Finally in
205   any boundary that combines a change in $k$ and a change in the ordering of the coordinates of $\boldsymbol{x}$ we
206   can combine the above arguments and prove that $\boldsymbol{f}$ is continous at these boundaries too.

207   **Output of $\boldsymbol{f}$ in $\Delta_{d-1}$.** We fix $k_{\boldsymbol{x}}$ to be $k$ and we consider without loss of generality a vector $\boldsymbol{x}$ that
208   satisfies

$$x_1 \geq x_2 \geq \cdots \geq x_d. \tag{B.12}$$

209   Therefore

$$\boldsymbol{f}(\boldsymbol{x}) = \frac{1}{\delta} \boldsymbol{SM}_{(k,d)} \cdot \boldsymbol{x} + \boldsymbol{u}_k.$$

210   From the definition of softmax matrices we have that for any column $\boldsymbol{m}_j$ of $\boldsymbol{SM}_{(k,d)}$, $\sum_{i=1}^{d} m_{ij} =$
211   $0$ and since $\sum_{i=1}^{d} u_{ki} = 1$ we have that for any $\boldsymbol{x} \in \mathbb{R}^d$, $\sum_{i=1}^{d} f_i(\boldsymbol{x}) = 1$. Hence it remains to
212   prove that $f_i(\boldsymbol{x}) \geq 0$.

213   Let $\boldsymbol{s}_i^T$ be the $i$th row of $\boldsymbol{SM}_{(k,d)}$. For $i > k$ we have $\boldsymbol{s}_i^T = \boldsymbol{0}^T$ and $u_{ki} = 0$, hence $f_i(\boldsymbol{x}) = 0$. On
214   the other hand, if $i \leq k$, we have that for

$$f_i(\boldsymbol{x}) = \frac{1}{\delta} \sum_{j=1}^{d} s_{ij} x_j + \frac{1}{k} = -\frac{1}{\delta k} x_1 + \frac{1}{\delta i} x_i + \frac{1}{\delta} \sum_{j=i+1}^{k} s_{ij} x_j + \frac{1}{k}$$

215   but for $j > i$ $s_{ij} \leq 0$ and because of (B.12) we have that

$$f_i(\boldsymbol{x}) \geq -\frac{1}{\delta k} x_1 + \frac{1}{\delta} \left( \frac{1}{i} + \sum_{j=i+1}^{k} s_{ij} \right) x_2 + \frac{1}{k} = -\frac{1}{\delta k} x_1 + \frac{1}{\delta} \left( \sum_{j=i}^{k} s_{ij} \right) x_2 = -\frac{1}{\delta k} (x_1 - x_2) + \frac{1}{k}$$

216   now by the definition of $k$ we have that $-(x_1 - x_2) \geq -\delta$ and hence

$$f_i(\boldsymbol{x}) \geq -\frac{1}{\delta k} \delta + \frac{1}{k} = 0.$$

217   This finishes the proof that $\boldsymbol{f}(\boldsymbol{x})$ is a probability distribution.

218   We are now ready to prove the two parts of Theorem 4.3.

219 **Proof of 1.** Without loss of generality we can again assume that $\boldsymbol{x}$ satisfies (B.12) and we again fix
220 $k = k_{\boldsymbol{x}}$. In this case the condition $\|\boldsymbol{x}\|_\infty - x_i > \delta$ translates to $i > k$. Then by the definition of $\boldsymbol{f}$
221 we have that

$$f_i(\boldsymbol{x}) = \boldsymbol{s}_i^T \boldsymbol{x} + u_{ki}$$

222 but by the definition of $\boldsymbol{SM}_{(k,d)}$ we have that $\boldsymbol{s}_i^T = \boldsymbol{0}^T$ and $u_{ki} = 0$. These two imply $f_i(\boldsymbol{x}) = 0$.

223 **Proof of 2.** Since $\boldsymbol{f}$ is continuous and piecewise linear we can use Lemma B.2 and we get

$$\|\boldsymbol{f}(\boldsymbol{x}) - \boldsymbol{f}(\boldsymbol{y})\|_q \leq \left( \max_{\boldsymbol{A} \in \mathcal{A}_{\boldsymbol{f}}} \|\boldsymbol{A}\|_{p,q} \right) \cdot \|\boldsymbol{x} - \boldsymbol{y}\|_p \quad \forall \boldsymbol{x}, \boldsymbol{y} \in \mathbb{R}^d.$$

224 Now we have that the set $\mathcal{A}_{\boldsymbol{f}}$ is the following

$$\mathcal{A}_{\boldsymbol{f}} = \left\{ \frac{1}{\delta} \boldsymbol{P}^T \boldsymbol{SM}_{(k,d)} \boldsymbol{P} \mid k \in \mathbb{N}, \boldsymbol{P} \text{ permutation matrix} \right\}$$

225 and since $\boldsymbol{P}$ is a permutation matrix we have that

$$\left\| \boldsymbol{P}^T \boldsymbol{SM}_{(k,d)} \boldsymbol{P} \right\|_{p,q} = \left\| \boldsymbol{SM}_{(k,d)} \right\|_{p,q}$$

226 which implies

$$\|\boldsymbol{f}(\boldsymbol{x}) - \boldsymbol{f}(\boldsymbol{y})\|_q \leq \frac{1}{\delta} \left( \max_{k \in [d]} \left\| \boldsymbol{SM}_{(k,d)} \right\|_{p,q} \right) \cdot \|\boldsymbol{x} - \boldsymbol{y}\|_p \quad \forall \boldsymbol{x}, \boldsymbol{y} \in \mathbb{R}^d.$$

227 Finally using Lemma B.8 we have that

$$\|\boldsymbol{f}(\boldsymbol{x}) - \boldsymbol{f}(\boldsymbol{y})\|_q \leq \frac{2 \min \left\{ \frac{q}{q-1}, p+1, \log d \right\}}{\delta} \|\boldsymbol{x} - \boldsymbol{y}\|_p \quad \forall \boldsymbol{x}, \boldsymbol{y} \in \mathbb{R}^d.$$

228 This completes the proof of the theorem.

## C Proofs of Lower Bounds in Section 4.2

230 In this section we provide the proofs of Theorem 4.4 and Theorem 4.5.

### C.1 Proof of Theorem 4.4

232 We will show our proof of all the dimensions $d$ of the form $d = 2^{2k}$, $k \in \mathbb{N}_+$. Then we can deduce
233 that asymptotically our lower bound holds. We use an induction argument with base case $d = 2$ and
234 inductive step from $d$ to $d^2$.

235 **Induction Base, $d = 2$.** In this case we have that $\boldsymbol{f}(\boldsymbol{x}) = (f_1(x_1, x_2), 1 - f_1(x_1, x_2))$ and for
236 simplicity we use the notation $f$ to refer to $f_1$. We will prove that the $\ell_\infty$ to $\ell_1$ Lipschitz constant
237 of $\boldsymbol{f}$ is at least $1/\delta$ even in the restricted subregion where $x_1 + x_2 = a$ for some $a \in \mathbb{R}_+$. In this
238 region the problem becomes single dimensional since $\boldsymbol{f}(\boldsymbol{x}) = (f_1(x_1, a - x_1), 1 - f_1(x_1, a - x_1))$
239 and the only freedom of $\boldsymbol{f}$ is to decide the single dimensional function $f(x) = f_1(x, a - x)$. The
240 approximation constraint implies that

$$\max\{x, a - x\} - x f(x) - (a - x)(1 - f(x)) \leq \delta \Leftrightarrow$$

241

$$\Leftrightarrow (a - 2x) f(x) \leq \delta - \max\{x, a - x\} + a - x.$$

242 The last inequality implies that there are two regions of $[0, a] \times [0, 1]$ where $(x, f(x))$ cannot be in.
243 The first is for $x \leq a/2$ where $(E_1) : f(x) \leq \delta/(a - 2x)$ and the second is for $x > a/2$ where
244 $(E_2) : f(x) \geq 1 + \delta/(a - 2x)$. Every $f$ that satisfies the approximation conditions has to avoid
245 the regions $(E_1)$ and $(E_2)$. Since we our goal is to minimize the Lipschitz constant of $\boldsymbol{f}$ in this one
246 dimensional projection of $\boldsymbol{f}$ we want to see what is the minimum $df/dx$ that we can achieve while
247 $f$ avoids $(E_1)$ and $(E_2)$ and it is defined in the whole interval $[0, a]$. The forbitten regions $(E_1)$ and
248 $(E_2)$ together with the optimal such $f$ are shown in the next figure.

249 In it is not difficult to see that the any function $f : [0, a] \to [0, 1]$ that avoids $(E_1)$ and $(E_2)$ has
250 to have at some point $\xi \in [0, a]$ a slope $f'(\xi)$ that is at least the slope of the green line in Figure 2

Figure 2: The forbitten regions $(E_1)$, $(E_2)$ and the optimal function $f$ for $a = 2$, $\delta = 1/10$.

which represents the line that is both targent to the boundary of $(E_1)$ and to the boundary of $(E_2)$. This target line can we computed in a closed form and its slope can be shown to be greater than $1/8\delta$. We leave the precise calculation as an exercise to the reader.

**Inductive Step, from $d$ to $d^2$.** We assume by inductive hypothesis that for any soft maximum function $\boldsymbol{f}$ in $d$ dimensions, with Lipschitz constant at most $\log(d)/8\delta$ has expected approximation loss at least $\delta$. We will then prove that for any soft maximum function $\boldsymbol{f}$ in $d^2$ dimensions with Lipschitz constant at most $\log(d)/8\delta$ has expected approximation loss at least $2 \cdot \delta$. This in turn implies that if $\boldsymbol{f}$ has Lipschitz constant at most $2\log(d)/8\delta$ then $\boldsymbol{f}$ has expected approximation loss at least $\delta$.

Consider any soft maximum function $\boldsymbol{f} : \mathbb{R}_+^{d^2} \to \Delta_{d^2-1}$ and let

$$\delta^* = \max_{\boldsymbol{z} \in \mathbb{R}_+^{d^2}} \|\boldsymbol{z}\|_\infty - \langle \boldsymbol{f}(\boldsymbol{z}), \boldsymbol{z} \rangle.$$

We restrict our attention to a subspace of $\mathbb{R}_+^{d^2}$ that is produced by $\left(\mathbb{R}_+^d\right)^2$ by the following map $\boldsymbol{g} : \left(\mathbb{R}_+^d\right)^2 \to \mathbb{R}_+^{d^2}$ defined as

$$g_\ell(\boldsymbol{x}, \boldsymbol{y}) = x_{\ell \bmod d} + y_{\ell \operatorname{div} d}.$$

We also define

$$\hat{\delta} = \max_{\boldsymbol{x}, \boldsymbol{y} \in \mathbb{R}_+^d} \|\boldsymbol{g}(\boldsymbol{x}, \boldsymbol{y})\|_\infty - \langle \boldsymbol{f}(\boldsymbol{g}(\boldsymbol{x}, \boldsymbol{y})), \boldsymbol{g}(\boldsymbol{x}, \boldsymbol{y}) \rangle.$$

On these instances of $\mathbb{R}^{d^2}$ we want to view the space of alternatives $[d^2]$ as a product space $[d] \otimes [d]$ and that's what the mapping $\boldsymbol{g}$ is capturing. We also want to view the output distribution as a product distribution over $[d] \otimes [d]$ but since we cannot assume independence we only define the marginal distributions of $\boldsymbol{f}(\boldsymbol{z})$ to the coordinates $\ell$ that have index with the same value $\ell \bmod d$, and the coordinates $\ell$ that have the same value $\ell \operatorname{div} d$. We will call $\boldsymbol{q} : \mathbb{R}_+^{d^2} \to \Delta_d$ the marginal distribution to the coordinates $\ell$ that have index with the same value $\ell \operatorname{div} d$ and $\boldsymbol{r} : \mathbb{R}_+^{d^2} \to \Delta_d$ the marginal distribution to the coordinates $\ell$ that have the same value $\ell \bmod d$. More formally

$$q_i(\boldsymbol{z}) = \sum_{j=1}^{d} f_{id+j}(\boldsymbol{z})$$

$$\text{and } r_j(\boldsymbol{z}) = \sum_{i=1}^{d} f_{id+j}(\boldsymbol{z}).$$

Now it is easy to observe that

$$\|\boldsymbol{g}(\boldsymbol{x}, \boldsymbol{y})\|_\infty = \|\boldsymbol{x}\|_\infty + \|\boldsymbol{y}\|_\infty$$
$$\text{and } \langle \boldsymbol{f}(\boldsymbol{g}(\boldsymbol{x}, \boldsymbol{y})), \boldsymbol{g}(\boldsymbol{x}, \boldsymbol{y}) \rangle = \langle \boldsymbol{q}(\boldsymbol{g}(\boldsymbol{x}, \boldsymbol{y})), \boldsymbol{x} \rangle + \langle \boldsymbol{r}(\boldsymbol{g}(\boldsymbol{x}, \boldsymbol{y})), \boldsymbol{y} \rangle.$$

272 Hence

$$\|g(x,y)\|_\infty - \langle f(g(x,y)), g(x,y)\rangle = \underbrace{\|x\|_\infty - \langle q(g(x,y)), x\rangle}_{\delta_1(x,y)} + \underbrace{\|y\|_\infty - \langle r(g(x,y)), y\rangle}_{\delta_2(x,y)}$$

273 We now define a continuous two game with the following players:

274      1. the first player picks a strategy $x \in \mathbb{R}^d$ and has utility function equal to $\delta_1(x,y)$, and

275      2. the second player picks a strategy $y \in \mathbb{R}^d$ and has utility function equal to $\delta_2(x,y)$.

276 It is easy to see that since $f$ is Lipschitz continuous, both $q$ and $r$ are continuous and this implies
277 that $\delta_1$ and $\delta_2$ are continuous. It is well known then from the theory of continuous games that there
278 exists a mixed Nash Equilibrium in the game that we described above [33]. This means that there
279 exists a pair of distributions $\mathcal{D}_x, \mathcal{D}_y$ in $\mathbb{R}^d$ such that

280      1. for every $x^\star$ in the support of $\mathcal{D}_x$ it holds that $x^\star = \operatorname{argmax}_{x \in \mathbb{R}^d} \mathbb{E}_{y \sim \mathcal{D}_y}[\delta_1(x,y)]$, and

281      2. for every $y^\star$ in the support of $\mathcal{D}_y$ it holds that $y^\star = \operatorname{argmax}_{y \in \mathbb{R}^d} \mathbb{E}_{x \sim \mathcal{D}_x}[\delta_2(x,y)]$.

282 Let us know define the following functions

283      • $\bar{q}(x) = \mathbb{E}_{y \sim \mathcal{D}_y}[q(g(x,y))]$,

284      • $\bar{r}(y) = \mathbb{E}_{x \sim \mathcal{D}_x}[r(g(x,y))]$,

285      • $\bar{\delta}_1(x) = \mathbb{E}_{y \sim \mathcal{D}_y}[\delta_1(x,y)] = \|x\|_\infty - \langle \bar{q}(x), x\rangle$, and

286      • $\bar{\delta}_2(y) = \mathbb{E}_{x \sim \mathcal{D}_x}[\delta_2(x,y)] = \|y\|_\infty - \langle \bar{r}(y), y\rangle$

287 where in the definition of the last two functions we have used the linearity of expectation. Form the
288 existence of the Nash Equilibrium in the aforementioned continuous game we have that

$$\mathbb{E}_{x \sim \mathcal{D}_x, y \sim \mathcal{D}_y}[\delta_1(x,y) + \delta_2(x,y)] = \max_{x \in \mathbb{R}^d}\{\bar{\delta}_1(x)\} + \max_{y \sim \mathbb{R}^d}\{\bar{\delta}_2(y)\}$$

289 which in turn implies the following

$$\max_{x,y \in \mathbb{R}^d}\{\delta_1(x,y) + \delta_2(x,y)\} \geq \max_{x \in \mathbb{R}^d}\{\bar{\delta}_1(x)\} + \max_{y \sim \mathbb{R}^d}\{\bar{\delta}_2(y)\}. \tag{C.1}$$

290 Next our goal is to relate the Lipschitzness of $f$ with the Lipschitzness of $\bar{q}$ and $\bar{r}$. Observe that

$$\|g(x,y) - g(x',y')\|_\infty = \|x - x'\|_\infty + \|y - y'\|_\infty \tag{C.2}$$
$$\|q(g(x,y)) - q(g(x',y'))\|_1 \leq \|f(g(x,y)) - f(g(x',y'))\|_1 \tag{C.3}$$
$$\|r(g(x,y)) - r(g(x',y'))\|_1 \leq \|f(g(x,y)) - f(g(x',y'))\|_1 \tag{C.4}$$

291 where the first equality follows from simple calculations and the second and third inequality follow
292 from the known fact that the total variation distance of a distribution is lower bounded by the total
293 variation of its marginals.

294 Now we remind that we have assumed that $f$ has $(\ell_\infty, \ell_1)$-Lipschitz constant that is at most $L =$
295 $\log(d)/8\delta$. Using the fact that the $\ell_1$ norm is a convex function and using the Jensen inequality we
296 have that

$$\|\bar{q}(x) - \bar{q}(x')\|_1 \leq \mathbb{E}_{y \sim \mathcal{D}_y}[\|q(x,y) - q(x',y)\|_1]$$
$$\leq \mathbb{E}_{y \sim \mathcal{D}_y}[\|f(g(x,y)) - f(g(x',y))\|_1] \leq L \cdot \|x - x'\|_\infty \tag{C.5}$$

297 where the first inequality is due to Jensen, the second inequality follows from (C.3) and the last
298 inequality follows from the $(\ell_\infty, \ell_1)$-Lipschitz constant of $f$ and (C.2). The same way we can prove
299 the following

$$\|\bar{r}(y) - \bar{r}(y')\|_1 \leq L \cdot \|y - y'\|_\infty. \tag{C.6}$$

It hence follows that both $\bar{q}$ and $\bar{r}$ are softmax functions in $d$ dimensions with Lipschitz constant at most $L = \log(d)/8\delta$. Hence from our inductive hypothesis we have that the approximation error of both $\bar{q}$, $\bar{r}$ is at least $\delta$, of more formally

$$\max_{\boldsymbol{x}\in\mathbb{R}^d} \bar{\delta}_1(\boldsymbol{x}) \geq \delta \quad \text{and} \quad \max_{\boldsymbol{y}\in\mathbb{R}^d} \bar{\delta}_2(\boldsymbol{y}) \geq \delta.$$

Now putting the above inequalities together with (C.1) we get that the approximation error of $\boldsymbol{f}$ is at least $2\delta$. Formally $\max_{\boldsymbol{x},\boldsymbol{y}\in\mathbb{R}^d} \{\delta_1(\boldsymbol{x},\boldsymbol{y}) + \delta_2(\boldsymbol{x},\boldsymbol{y})\} \geq 2\delta$. This concludes the inductive step and proves our theorem.

## C.2 Proof of Theorem 4.5

We set $\boldsymbol{x} = (x, 0, \dots, 0)^T$ and $\boldsymbol{y} = (y, 0, \dots, 0)^T$, with $y > x$. Then we have

$$\text{EXP}(\boldsymbol{x}) = \left( \frac{e^{\alpha x}}{e^{\alpha x} + (d-1)}, \frac{1}{e^{\alpha x} + (d-1)}, \cdots, \frac{1}{e^{\alpha x} + (d-1)} \right)^T$$

$$\text{EXP}(\boldsymbol{y}) = \left( \frac{e^{\alpha y}}{e^{\alpha x} + (d-1)}, \frac{1}{e^{\alpha y} + (d-1)}, \cdots, \frac{1}{e^{\alpha y} + (d-1)} \right)^T.$$

Since $y > x$, we compute

$$\|\text{EXP}(\boldsymbol{x}) - \text{EXP}(\boldsymbol{y})\|_1 = \left( \frac{e^{\alpha y}}{e^{\alpha y} + (d-1)} - \frac{e^{\alpha x}}{e^{\alpha x} + (d-1)} \right)$$
$$- (d-1)\left( \frac{1}{e^{\alpha y} + (d-1)} - \frac{1}{e^{\alpha x} + (d-1)} \right)$$

and $\|\boldsymbol{x} - \boldsymbol{y}\|_p = y - x$. Now let

$$h(z) = \frac{e^{\alpha z}}{e^{\alpha z} + (d-1)} - (d-1)\frac{1}{e^{\alpha y} + (d-1)} = \frac{e^{\alpha z} - (d-1)}{e^{\alpha z} + (d-1)}$$

our goal to maximize, with respect to $x, y \in \mathbb{R}_+$ with $y \geq x$, the ratio

$$\frac{\|\text{EXP}(\boldsymbol{x}) - \text{EXP}(\boldsymbol{y})\|_1}{\|\boldsymbol{x} - \boldsymbol{y}\|_p} = \frac{h(y) - h(x)}{y - x}.$$

Because of the mean value theorem this is equivalent with maximum with respect to $z \in \mathbb{R}_+$ the derivative of $h$, $h'(z)$. But we have

$$h'(z) = \frac{\alpha e^{\alpha z}(e^{\alpha z} + (d-1)) - \alpha e^{\alpha z}(e^{\alpha z} - (d-1))}{(e^{\alpha z} + (d-1))^2} = 2\alpha\frac{e^{\alpha z}(d-1)}{(e^{\alpha z} + (d-1))^2}.$$

Now we set $z = \frac{\log d}{\alpha}$ and we get for $d > 10$

$$h'\left( \frac{\log d}{\alpha} \right) = 2\alpha\frac{d(d-1)}{(2d-1)^2} \leq \frac{\alpha}{2}.$$

Finally since the absolute approximation error of the exponential mechanism with parameter $\alpha$ is $\log d/\alpha$, to get $\delta$ absolute error we have to set $\alpha = \log d/\delta$ and hence for this regime

$$c \geq \frac{\log d}{2\delta}$$

and the proof of the theorem is completed.

# D  Application to Mechanism Design

In this section we show how to design a digital auction with limited supply and worst case guarantees. As we will see to do so we need to relax the incentive compatibility constraints to approximate incentive compatibility in the framework as in [22]. In this setting we fix an anonymous price for

all the agents regardless of whether their values follow the same distribution of not. In this case we
show that we can extract almost the optimal revenue among all the fixed price auctions.

Compared to the results of [22] and [1] our mechanism can interpolate between both of the results.
Most importantly our results, in contrast to both [22] and [1] achieves a worst case guarantee instead
of a guarantee in expectation or with high probability. Another improvement of our result is that it
holds even if we do not assume unlimited supply but we only have finite supply of the item to sell.

We start with the next Section D.1 with the basic definitions and formulation of the mechanism and
auction design problem.

## D.1 Definitions and Preliminaries

We first give some necessary basic definitions of design auctions for selling $k$ identical items to $n$
independent bidders with unit demand valuations.

**Items.** We have $k$ identical items for sell.

**Bidders.** We have $n$ independent bidders with unit demand valuations over the $k$ item to sell. The
bidders are clustered in $t$ classes and let $t(i)$ be the class of bidder $i$. The value of bidder $i \in [n]$
for any of the items is $v_i \in [0, H]$ where $H$ is the maximum possible value that we assume to be
known. We also assume that $v_i$ it is drawn from a distribution $\mathcal{F}_{t(i)}$. We assume that all the random
variables $v_i$ are independent from each other.

**Mechanism.** A mechanism $M$ is a function $M : \mathbb{R}_+^n \to \Delta_n^k \times \mathbb{R}_+^n$ that takes as input the bid
of the players and outputs $k$ probability distributions $\boldsymbol{A} = (\boldsymbol{a}_1, \ldots, \boldsymbol{a}_k) \in \Delta_n^k$ over the bidders
that determines the probability that each bidder is going to receive the item $j$, together with a non-
negative value $p_i$ for every bidder $i$ that determines the money bidder $i$ will pay. We write $M(\boldsymbol{v}) =$
$(\boldsymbol{A}, \boldsymbol{p})$ and we call $\boldsymbol{A} \in \Delta_n^k$ the allocation rule of the mechanism $M$ and $\boldsymbol{p}$ the payment rule of $M$.

**Bidders Utility.** We assume that the bidders are unit-demand and they have quasi-linear utility, i.e.
that the utility function $u_i : \Delta_n \times \mathbb{R}_+^n \to \mathbb{R}$ of each bidder is equal to $u_i(\boldsymbol{A}, \boldsymbol{p}) = \max_j (a_{ij} v_i) - p_i$.

**Revenue Objective.** For every mechanism $M$ the revenue $\text{REV}(M, \boldsymbol{v})$ the designer gets in input $\boldsymbol{v}$
is equal to $\text{REV}(M, \boldsymbol{v}) = \sum_{i \in [n]} p_i$ where $\boldsymbol{p}$ is the vector of prices that the mechanism $M$ assigns
to the agents in input $\boldsymbol{v}$. By $\text{REV}(M)$ we denote the expected value of the mechanism $M$ when the
values $\boldsymbol{v}$ are drawn from their distributions, i.e. $\text{REV}(M) = \mathbb{E}[\text{REV}(M, \boldsymbol{v})]$.

**Incentive Compatibility.** A mechanism $M$ is called *dominant strategy incentive compatible* (DSIC)
or simply *incentive compatible* (IC) if the bidders cannot increase their revenue by misreporting their
bids. More precisely we say that $M$ satisfies incentive compatibility if for every bidder $i$

$$u_i(M(v_i, \boldsymbol{v}_{-i})) \geq u_i(M(v_i', \boldsymbol{v}_{-i})) \qquad \forall\, v_i, v_i', \boldsymbol{v}_{-i}. \tag{D.1}$$

Also we say that $M$ is *$\varepsilon$-incentive compatible if for every bidder $i$*

$$u_i(M(v_i, \boldsymbol{v}_{-i})) \geq \cdot u_i(M(v_i', \boldsymbol{v}_{-i})) - \varepsilon \qquad \forall\, v_i, v_i', \boldsymbol{v}_{-i}. \tag{D.2}$$

**Individual Rationality.** We say that a mechanism $M$ satisfies individual rationality if for every
bidder $i$ $u_i(M(\boldsymbol{v})) \geq 0$ for all $\boldsymbol{v} \in \mathbb{R}_+^n$.

**Optimal Revenue over a Ground Set.** Let $\mathcal{M} = \{M_1, \ldots, M_d\}$ be a set of mechanisms which
we call *ground set*, we define the maximum revenue of $\mathcal{M}$ at input $\boldsymbol{v}$ as $\text{OPTREV}(\mathcal{M}, \boldsymbol{v}) =$
$\max_{M \in \mathcal{M}} \text{REV}(M, \boldsymbol{v})$. Also we define maximum expected revenue achievable by any mechanism
in $\mathcal{M}$ to be $\text{OPTREV}(\mathcal{M}) = \max_{M \in \mathcal{M}} \text{REV}(M)$.

The mechanisms that we describe in this section involve a smooth selection of a mechanism among
the mechanisms in a carefully chosen ground set of incentive compatible and individual rational
mechanisms $\mathcal{M}$.

**Soft Maximizer Mechanism.** Let $\mathcal{M} = \{M_1, \ldots, M_d\}$ be a ground set of incentive compatible
and individually rational mechanism. We define the mechanism $Q[\mathcal{M}, \boldsymbol{f}]$ to be the mechanism that
chooses one of the mechanisms in $[d]$ randomly from the probability distribution that output the soft
maximum function $\boldsymbol{f}$ with input the vector $\boldsymbol{x} = (\text{REV}(M_1, \boldsymbol{v}), \ldots, \text{REV}(M_d, \boldsymbol{v}))$.

The following lemma proves the incentive compatibility properties of the mechanism $Q[\mathcal{M}, \boldsymbol{f}]$ when the $\boldsymbol{f}$ satisfies some stability properties. For a proof of this lemma we refer to the proof of Lemma 3 in McSherry and Talwar [22].

**Lemma D.1.** *Let the bidders valuations come from the interval $[0, H]$, let also $\mathcal{M} = \{M_1, \ldots, M_d\}$ be a ground set of incentive compatible and individually rational mechanism and $\boldsymbol{f}$ be a soft maximum function that is $(\ell_p, \ell_1)$-Lipschitz with Lipschitz constant $L = \varepsilon/S_\chi(\text{REV})$. Then the mechanism $Q[\mathcal{M}, \boldsymbol{f}]$ is individually rational and $\varepsilon$-incentive compatible.*

## D.2 Selling Digital Goods with Anonymous Price

The single parameter auctions are arguably the most classical setting in the mechanism design literature. Myerson, in his seminal work [27], proved that among all the possible auction designs the revenue is maximized by a second price auction with reserve price. The basic assumptions of his framework though is the assumption that the auctioneer has a prior belief for the values of the different bidders and she tries to maximize her expected revenue in this Bayesian setting. This assumption is the major milestone in applying the Myerson's auction in practice. Trying to relax this assumption, a line of theoretical computer science work studied the maximization of revenue when we only have access to samples that come from the bidders distribution and not access to the entire distribution [29, 9, 6, 25, 8, 5]. Although these works make a very good progress on understanding the optimal auctions and make them more practical there are still some drawbacks that make these auctions not applicable in practice.

1. **Buyers may strategize in the collection of samples.** If the buyers know that the seller is going to collect samples to estimate the optimal auction to run then they have incentives to strategize so that the seller chooses lower prices and hence they get more utility.

2. **Constant approximation is not always a satisfying guarantee.** The constant approximation is a worst case guarantee and hence the constant approximation mechanisms might fail to get almost optimal revenue even in the instances where this is easy. A popular alternative in practical applications of mechanism design is to choose the optimal from a set of simple mechanisms.

Because of these reasons, 1. and 2., the implementation and the theoretical guarantees of the mechanism $Q[\mathcal{M}, \boldsymbol{f}]$ becomes a relevant problem. The ground set of mechanisms that we consider in this section is a subset of the second price selling separately auctions with a single reserved price, which we call set of anonymous auctions and we denote by $\mathcal{M}_A$. We are now ready to prove the main result of this section.

**Theorem D.2.** *Consider a $k$ identical item auction instance with unit demand bidder's and values in the range $[0, H]$. Then there exists a ground set of mechanisms $\hat{\mathcal{M}} \subseteq \mathcal{M}_A$ such that for all $\boldsymbol{v} \in [0, H]^n$ and for any of the possible outputs of $Q\left[\hat{\mathcal{M}}, \text{PLSOFTMAX}^\eta\right]$ with input $\boldsymbol{v}$ it holds that*

$$\text{REV}(Q\left[\hat{\mathcal{M}}, \text{PLSOFTMAX}^\eta\right], \boldsymbol{v}) \geq (1-\delta)\text{OPTREV}(\mathcal{M}_A, \boldsymbol{v}) - 4\left(\frac{1}{\delta} - 1\right)\frac{H}{\varepsilon}$$

*where $\text{PLSOFTMAX}^\eta$ is the soft maximum function defined in (4.1) with parameter such that $\text{PLSOFTMAX}$ is $\varepsilon$-Lipschitz in Total Variation Distance. Moreover $Q[\hat{\mathcal{M}}, \text{PLSOFTMAX}]$ is individually rational and $\varepsilon \cdot H$-incentive compatible.*

*Proof.* Let $[0, H]$ be the range of prices for the single item auction. We fix a positive real number $\delta$ and we use the discretization $\mathcal{P}$ of $[0, H]$, where $\mathcal{P} = \{p_1, \ldots, p_d\}$ and $p_i = H \cdot (1-\delta)^i$. Let also $\alpha = p_d$. We are now ready to define the ground set of mechanisms $\hat{\mathcal{M}} = \{M_1, \ldots, M_d\}$ where $M_i$ is the second price auction with reserved price equal to $p_i$. The size of $\hat{\mathcal{M}}$ is

$$d = \log\left(\frac{\alpha}{H}\right) / \log(1-\delta) \leq 2\log\left(\frac{H}{\alpha}\right)/\delta$$

where the last inequality follows assuming that $\delta \leq 1/2$. As we described, we will run our mechanism PLSOFTMAX, with objective function REV. In order to be able to apply our main

theorem about the PLSOFTMAX mechanism we will bound the $\ell_1$-sensitivity of the vector $\boldsymbol{x} = (\text{REV}(M_1, \boldsymbol{v}), \ldots, \text{REV}(M_d, \boldsymbol{v}))$ with respect the change of the bid of one agent. Hence we need to bound the quantity

$$\sum_{i=1}^{d} |\text{REV}(M_i, (v_i, \boldsymbol{v}_{-i})) - \text{REV}(M_i, (v_i', \boldsymbol{v}_{-i}))| \leq (1 - \delta) \frac{H}{\delta}.$$

This inequality holds because for every agent $i$ the total change that agent $i$ can make in the revenue objective of all the alternatives is at most

$$\sum_{i=1}^{d} (1 - \delta)^i H \leq \left( \frac{1}{\delta} - 1 \right) H,$$

which implies that for our setting $S_1(\text{REV}) \leq \left( \frac{1}{\delta} - 1 \right) H$.

The approximation loss of our mechanism has three components: (1) we loose $\delta\text{OPT}$ because of the discretization of the price of every item, (2) we loose $\alpha$ from every item because we need the ground set to be finite and (3) we loose $\eta$ because we use the soft maximization algorithm PLSOFTMAX$^\eta$. For the last part and since we need PLSOFTMAX$^\eta$ to be $\varepsilon$-Lipschitz in total variation distance we have that

$$\varepsilon = \frac{4}{\eta} S_1(\text{REV}) \leq \frac{4}{\eta} \left( \frac{1}{\delta} - 1 \right) H \implies \eta \leq \frac{4}{\varepsilon} \left( \frac{1}{\delta} - 1 \right) H.$$

Finally applying Theorem 4.3 the theorem follows. $\qquad\square$

If we assume that $H = O(1)$ then by setting $\delta \leftarrow \frac{1}{\sqrt{\text{OPT}}}$ and $\varepsilon \leftarrow \varepsilon \cdot H$ we recover the result of [1], with relaxed incentive compatibility, but even in the case of limited supply and having a worst case guarantee.

**Corollary D.3.** *Consider a $k$ identical item auction instance with unit demand bidder's and values in the range $[0, H]$. If we fix $H$ then there exists a mechanism $M$ such that for any $\boldsymbol{v} \in [0, H]^n$, for all $\boldsymbol{v} \in [0, H]^n$ and for any of the possible outputs of $M$ with input $\boldsymbol{v}$ it holds that*

$$\text{REV}(M, \boldsymbol{v}) \geq \text{OPTREV}(\mathcal{M}_A) - O \left( \frac{1}{\varepsilon} \sqrt{\text{OPTREV}(\mathcal{M}_A)} \right)$$

*where $M$ is individually rational and $\varepsilon$-incentive compatible.*

Another corollary can be directly derived by applying a discretized version of the Theorem 9 of [22] but replacing the exponential mechanism with the PLSOFTMAX mechanism. Then as we explained in Section 4 the guarantees will hold in the worst case and not in expectation.

**Corollary D.4.** *Consider a $k$ identical item auction instance with unit demand bidder's and values in the range $[0, H]$. If we fix $H$ then there exists a mechanism $M$ such that for any $\boldsymbol{v} \in [0, H]^n$, for all $\boldsymbol{v} \in [0, H]^n$ and for any of the possible outputs of $M$ with input $\boldsymbol{v}$ it holds that*

$$\text{REV}(M, \boldsymbol{v}) \geq \text{OPTREV}(\mathcal{M}_A) - O \left( \frac{1}{\varepsilon} \log \left( \text{OPTREV}(\mathcal{M}_A) \cdot k \right) \right)$$

*where $M$ is individually rational and $\varepsilon$-incentive compatible.*

As we can see Corollary D.3 and Corollary D.4 are not directly comparable since in Corollary D.4 the $\log(k)$ factor in the approximation error appears that misses from Corollary D.3.

# E   Maximization of Submodular Functions

In this section we consider the problem of differential privately maximizing a submodular function, under cardinality constraints. For this problem we apply the power mechanism and we compare our results with the state of the art work of Mitrovic et al. [24]. We observe that when the input data set is only $O(1)$-multiplicative insensitive power mechanism has an error that is asymptotically smaller than the corresponding error from the state of the art algorithm of Mitrovic et al. [24]. This result is formally stated in Corollary 6.6.

As discussed in Section 6.2.1, to solve the submodular maximization under cardinality constraints we use the Algorithm 1 of [24], where we replace the exponential mechanism in the soft maximization step with the power mechanism.

**Algorithm 1** (Algorithm 1 of [24]):

**Input:** submodular function $h$, soft maximization function $g$, $k \in \mathbb{N}$.

**Output:** $S \subseteq \mathcal{D}$ such that $|S| = k$.

1. Initialize $S_o = \emptyset$. Let $|\mathcal{D}| = d$ and $\mathcal{D} = \{v_1, \ldots, v_d\}$.
2. For $i \in [k]$:
   a. Define $q_i : \mathcal{D} \setminus S_{i-1} \to \mathbb{R}$ as
   $$q_i(v) = h(S_{i-1} \cup \{v\}) - f(S_{i-1}).$$
   b. Pick $u_i \in \mathcal{D}$ from the probability distribution
   $$g\left(q_i(v_1), \ldots, q_i(v_d)\right).$$
   c. $S_i \leftarrow S_i \cup \{u_i\}$.
3. Return $S_k$.

To analyze Algorithm 1 we need the following result for compositions of differentially private algorithms.

**Composition of Differentially Private Algorithms.** An algorithm $A$ is a composition of $k$ algorithms $A_1, \ldots, A_k$ if the output of $A(v)$ is a function only of the outputs $A_1(v), \ldots, A_k(v)$.

The following theorem bounds the privacy of $A(v)$ as a function of the privacy of $A_1(v), \ldots, A_k(v)$.

**Theorem E.1** ([12]). *Let $A_1, \ldots, A_k$ be differentially private algorithms with parameters $(\varepsilon', \delta')$. Let also $A$ a composition of $A_1, \ldots, A_k$. Then, $A$ satisfies $(\varepsilon, \delta)$-differential privacy with*

1. $\varepsilon = k\varepsilon'$ and $\delta = k\delta'$,
2. $\varepsilon = \frac{1}{2}k^2\varepsilon'^2 + \sqrt{2\log(1/\eta)}\varepsilon'$ and $\delta = \eta + k\delta'$ for any $\eta > 0$.

We are now ready to prove Theorem 6.4.

*Proof of Theorem 6.4.* The privacy guarantee easily follows from the composition properties of differentially private mechanisms that we present in Theorem E.1.

Let $S^*$ be the set of the optimal solution, $S_i$ be the set that the algorithm has in the $i$th iteration and $v_i$ the $i$th element that our algorithm chose. We have that
$$\mathbb{E}[h(S_i \cup \{v_i\}) - h(S_i)] =$$
$$= \frac{1}{1+\delta} \max_{v \in \mathcal{D} \setminus S_{i-1}} (h(S_i \cup \{v\}) - h(S_i))$$
$$\geq \frac{1}{1+\delta} \frac{1}{k} \left( \sum_{v \in S^*} (h(S_i \cup \{v\}) - h(S_i)) \right)$$
$$\geq \frac{1}{1+\delta} \frac{1}{k} (h(S^* \cup S_{i-1}) - h(S_{i-1}))$$
$$\geq \frac{1}{1+\delta} \frac{1}{k} (\text{OPT} - h(S_{i-1})).$$

Therefore
$$\text{OPT} - \mathbb{E}[h(S_i)] \leq \left( 1 - \frac{1}{1+\delta} \frac{1}{k} \right)^i \text{OPT}.$$

From which we conclude
$$\mathbb{E}[h(S_k)] \geq \left( 1 - \left( 1 - \frac{1}{1+\delta} \frac{1}{k} \right)^k \right) \text{OPT}$$
$$\geq \left( 1 - \frac{1}{\exp(1/(1+\delta))} \right) \text{OPT}.$$

474 and hence the theorem follows. □

475 Next our goal is to compare Theorem 8 of [24] with Theorem 6.4. We illustrate the difference be-
476 tween power and exponential mechanism showing an improvement over the state of the art algorithm
477 of [24].

478 **Lemma E.2.** *Let $\delta_{\mathrm{POW}}$ be the approximation loss of* POW *assuming that the input data set is $t$-*
479 *multiplicative insensitive, then $\delta_{\mathrm{POW}} \leq \min\left\{ \frac{1}{e} + \frac{2\sqrt{k}\log d}{t\varepsilon}\frac{S_\infty(h)}{\mathrm{OPT}}, 1 \right\}$.*

480 *Proof.* From Theorem 6.4 we have that

$$
\delta_{\mathrm{POW}} = \min\left\{ \exp\left( -\left(1 - \frac{S_\infty(h)}{\mathrm{OPT}}\right)^{\frac{2\sqrt{k}\log d}{\varepsilon}} \right), 1 \right\}
$$

$$
\leq \min\left\{ \exp\left( -\left(1 - \frac{2\sqrt{k}\log d}{\varepsilon}\frac{S_\infty(h)}{\mathrm{OPT}}\right) \right), 1 \right\}
$$

$$
= \frac{1}{e}\min\left\{ \exp\left( \frac{2\sqrt{k}\log d}{\varepsilon}\frac{S_\infty(h)}{\mathrm{OPT}} \right), e \right\}
$$

481 Now if $\frac{2\sqrt{k}\log d}{\varepsilon}\frac{S_\infty(h)}{\mathrm{OPT}} \geq 1$ then $\delta_{\mathrm{POW}} = 1$ and hence, we can assume that $\frac{2\sqrt{k}\log d}{\varepsilon}\frac{S_\infty(h)}{\mathrm{OPT}} \leq 1$. But
482 for any $z \leq 1$ it is easy to see that $e^z \leq 1 + ez$ and hence

$$
\delta_{\mathrm{POW}} \leq \frac{1}{e}\min\left\{ 1 + e\frac{2\sqrt{k}\log d}{\varepsilon}\frac{S_\infty(h)}{\mathrm{OPT}}, e \right\}
$$

483 and the lemma follows. □

484 Now combining Theorem 6.4 and Lemma E.2 we can prove Corollary 6.6 which clearly illustrates
485 the comparison of the performance of power and exponential mechanism. From Corollary 6.6 we
486 observe that the approximation loss using the exponential mechanism is a $O(\sqrt{k})$ factor larger than
487 the approximation loss using the power mechanism. Hence Corollary 6.6 improves over the state of
488 the art differentially private algorithms for submodular optimization.

489 We can use the same ideas as in Theorem 6.4 and Corollary 6.6 to improve the results for maximiza-
490 tion of submodular functions with more general matroid constraints of [24].

## F   Experiments on Large Real-World Data Sets

492 ***Remark.*** *In the main part we accidentally refer to Appendix F both for the theoretical and the*
493 *practical results about differentially private submodular maximization. Please look at the Appendix*
494 *E for the details on the theoretical part and in this section for the details in the experiments part.*

495 We now empirically validate our results for submodular maximization. In our experiments we used
496 a publicly available data-set to create a max-k-coverage instance similarly to prior work [13]. In a
497 coverage instance we are given a family $N$ of sets over a ground set $U$ and we want to find $k$ sets
498 from $N$ with maximum size of their union (which is a monotone submodular maximization prob-
499 lem under cardinality constraint). We created the coverage instance from the **DBLP co-authorship**
500 network of computer scientists by extracting, for each author, the set of her coauthors. The ground
501 set is the set of all authors in DBLP. There are $\sim 300$ thousands sets over $\sim 300$ thousands elements
502 for a total sum of sizes of all sets of $1.0$ million. Then we ran the (non-private) greedy submodular
503 maximization algorithm to obtain a (baseline) upperbound on the solution (notice that computing
504 the actual optimum is NP-Hard). Then we compared the objective value obtained by private greedy
505 algorithm for submodular maximization using the exponential mechanism (as described in Algo-
506 rithm 1 in [24]) and using the power mechanism as soft-max, for different values of the parameter $\alpha$
507 in the two methods. We used $k = 10$ as the cardinality of the output in our experiment.

508 To evaluate empirically the smoothness of the mechanism we performed a manipulation test on the
509 data. We manipulated the coverage instance removing, independently, each element of the ground

Figure 3: Robustness vs objective value in the submodular maximization with cardinality constraint $k = 10$. The y-axis shows the ration of the average objective obtained vs the (non-private) greedy algorithm. The x-axis represent the sensitivity to the manipulation test of the value of the first element selected.

set with probability $1/1000$. Then, for a fixed mechanism and parameter setting, we compared the probability distribution of the first set selected by the algorithm in the manipulated instance vs in the original instance (we used the $\ell_1$ and $\ell_\infty$ distance of the distributions)[2]. Finally, we ran each configuration of the experiment (i.e., a mechanism and a parameter) 100 times and reported the average objective in the original dataset (over the objective of non-private greedy) and average distance between the distributions obtained over the original and manipulated datasets. Figures 3a and 3b report the results for $k = 10$ in the DBLP instance. Notice that we observe that for the same level of sensitivity to manipulation (both in $l_1$ and $l_\infty$ norm) the power mechanism obtains significantly more objective value in this problem as well (y-axis reports the average ratio of the objective obtained vs that of the non-private algorithm). This confirms our theoretical results for submodular maximization.

# G Loss Function For Multi-class Classification

Before presenting our loss function that can be used for multi-class classification we present a proof of Lemma 6.7. Due to a minor typo in the presentation of the Lemma in the main part of the paper we restate the Lemma here corrected.

**Lemma G.1** (Lemma 6.7). *Let* $h(\cdot) = \text{sparsegen-lin}(\cdot)$ *be the generalization of* $\text{sparsemax}(\cdot)$ *function, then there exist* $\boldsymbol{x}, \boldsymbol{y} \in \mathbb{R}^d$ *such that* $\|h(\boldsymbol{x}) - h(\boldsymbol{y})\|_1 \geq \frac{1}{2} d^{1-1/q} \|\boldsymbol{x} - \boldsymbol{y}\|_q$.

*Proof of Lemma 6.7.* We set $\boldsymbol{x} = \boldsymbol{0}$ and $\boldsymbol{y}$ such that $y_i = 2/d$ for $i \leq d/2$ and $y_i = 0$ otherwise. Doing simple calculations we get that $h(\boldsymbol{x}) = (1/d) \cdot \boldsymbol{1}$, whereas $h_i(\boldsymbol{y}) = 2/d$ for $i \leq d/2$ and $h_i(\boldsymbol{y}) = 0$ otherwise. Hence we have $\|h(\boldsymbol{x}) - h(\boldsymbol{y})\|_1 = 1$ and

$$\|\boldsymbol{x} - \boldsymbol{y}\|_q = (2/d)^{1-1/q} \leq 2/d^{1-1/q}$$

and the lemma follows. $\qquad\square$

In this section, we show how our mechanism can be used in multi-class classification by proposing the corresponding loss function.

First, we note that the $\mathcal{L}_{\text{sparsegen-lin,hinge}}$ loss function defined in [21] can be used as a loss function for any soft-max function that satisfies: (1) permutation invariance, (2) $\delta$-worst-case approximation additive loss, where we have to set $\delta = 1 - \lambda$. The main issue of this loss function is that it does not take into account specific structural properties of the softmax function used. For this reason, we propose an alternative loss function.

A loss function that corresponds to PLSOFTMAX with parameter $\delta$ is a function $L : \mathbb{R}^d \times \Delta_d \to \mathbb{R}_+$ such that for any $\boldsymbol{x} \in \mathbb{R}^d$ and $\boldsymbol{q} \in \Delta_d$, it holds that $L(\boldsymbol{x}; \boldsymbol{q}) = 0 \Leftrightarrow \text{PLSOFTMAX}^\delta(\boldsymbol{x}) = \boldsymbol{q}$. Our loss function has three components: (1) $L_{ord}$ is minimized only when the ordering of $\boldsymbol{x}$ is the same as the ordering of $\boldsymbol{q}$, (2) $L_{supp}$ is minimized when the coordinates of $\boldsymbol{x}$ that are within $\delta$ from $\|\boldsymbol{x}\|_\infty$ correspond to the coordinates $i$ such that $q_i > 0$, and (3) $L_{sqr}$ minimizes the error between $\text{PLSOFTMAX}^\delta(\boldsymbol{x})$ and $\boldsymbol{q}$ assuming they have the same order. Finally, our loss function $L_{\text{PLSOFTMAX}}$ is the sum of these three components, i.e. $L_{\text{PLSOFTMAX}} = L_{ord} + L_{supp} + L_{sqr}$.

**Order Regularization.** For every $\boldsymbol{q} \in \Delta_d$, let $\pi_{\boldsymbol{q}}$ be the permutation of the coordinates $[d]$ such that $q_{\pi_{\boldsymbol{q}}(1)} \geq \cdots \geq q_{\pi_{\boldsymbol{q}}(d)}$, then

$$L_{ord}(\boldsymbol{x}; \boldsymbol{q}) = \sum_{i=1}^{d-1} \max\{x_{\pi_{\boldsymbol{q}}(i+1)} - x_{\pi_{\boldsymbol{q}}(i+1)}, 0\}.$$

**Support Regularization.** Let $\boldsymbol{q} \in \Delta_d$, let $S \subseteq [d]$ be the subset of the coordinates $[d]$ such that $i \in S \Leftrightarrow q_i > 0$, let also $\delta$ be the parameter of PLSOFTMAX, then

$$L_{supp}(\boldsymbol{x}; \boldsymbol{q}) = \sum_{i \in S} \max\{x_{\pi_{\boldsymbol{q}}(1)} - x_i - \delta, 0\} + \sum_{i \in [d]\setminus S} \max\{x_i - x_{\pi_{\boldsymbol{q}}(1)} + \delta, 0\}.$$

**Square Loss.** Let $\boldsymbol{q} \in \Delta_{d-1}$, then

$$L_{sqr}(\boldsymbol{x}; \boldsymbol{q}) = \left\| \boldsymbol{q} - \frac{1}{\delta} \boldsymbol{P}_{\pi_{\boldsymbol{q}}}^{-1} \boldsymbol{S} \boldsymbol{M}_{(k_{\boldsymbol{q}}, d)} \boldsymbol{P}_{\pi_{\boldsymbol{q}}} \boldsymbol{x} - \boldsymbol{P}_{\pi_{\boldsymbol{q}}}^{-1} \boldsymbol{u}^{(k_{\boldsymbol{q}})} \right\|_2^2.$$

The main properties of the loss function $L_{\text{PLSOFTMAX}}$ are summarized in Proposition 6.8. This proposition suggests that $L_{\text{PLSOFTMAX}}$ can be used as a meaningful loss function in multiclass classification.

*Proof of Proposition 6.8.* The property (1) follows directly from the fact that $L_{\text{PLSOFTMAX}}$ is a sum of non-negative terms. Also observe that: (i) $L_{ord} = 0$ if and only if the order of the coordinates of the vector $\boldsymbol{x}$ agrees with the order of the coordinates of $\boldsymbol{q}$, and (ii) $L_{supp} = 0$ if and only if the only coordinates that are $\delta$-close to $\|\boldsymbol{x}\|_\infty$ are the coordinates for which $q_i > 0$. Using (i) and (ii) together with $L_{sqr} = 0$ we can see that the property (2) of Proposition 6.8 is implied. Property (3) follows again easily from the fact that the maximum of two convex function is convex and the sum of convex functions is also convex. $\qquad\square$

## Footnotes

[2]Ideally one would like to compare the distribution of the output value of the algorithm for the actual $k$. However, computing or even approximating well the distribution of value of the output is computationally hard, so we resort to computing exactly the distribution of the first item selected.