[Reviews · NeurIPS 2020]

Review 1

Summary and Contributions: A soft-max function has two main efficiency measures, approximation and smoothness. Authors goal is to identify the optimal approximation-smoothness tradeoffs for different measures of approximation and smoothness. They introduce a soft-max function, called piece-wise linear soft-max, with optimal tradeoff between approximation measured in terms of worst-case additive approximation, and smoothness measured with respect to l𝑞 -norm. The worst-case approximation guarantee of the piece-wise linear mechanism enforces sparsity in the output of our soft-max function, a property that is known to be important in Machine Learning applications and is not satisfied by the exponential mechanism. Finally, they investigate another soft-max function, called power mechanism, with optimal tradeoff between expected multiplicative approximation and smoothness with respect to the Rényi Divergence, which provides improved theoretical and practical results in differentially private submodular optimization.

Strengths: The PLSOFTMAX in Section 4 is quite interesting. Since it's a piece-wise linear function and achieves the best Lipschitz constant. This result might inspire us to seek some piece-wise linear function rather than smooth function when facing similar situation. During the proof, this is another interesting thing, the norm $l_r$ in Theorem B.6 is the dual norm of norm $l_p$. The dual norm also appears in Theorem 4.4 for norms $l_\infty$ and $l_1$. From the applications in Section 6, we can see that the soft-max function could be used to different areas. That stands out the meaning of `piece-wise linear' of the function PLSOFTMAX, since it gives us another idea to find proper functions when dealing with different situations.

Weaknesses: The result is quite good.

Correctness: For Theorem 3.2, from the proof we can see that $c$ could be equal to $ \frac{\log d-2}{2}$, then the result in the statement of Theorem 3.2 should be $c\ge \frac{\log d-2}{2}$ In the Appendix B, for the definition of $\bm{m}_i^{(k,d)}$ and $\bm{s}_i^{(k,d)}$, using words to describe is enough. The dashes in illustration (B.6) and (B.7) are redundant. In line 173(Appendix), the Riemann zeta function should be an infinite sum rather than the finite sum appears here. Therefore, line 174 should be edited(this will not influence the result).

Clarity: Main paper: Both `soft-max function' and `softmax function' (in Section 1) appear. Renyi, line 48. $y \to \bm{y}$, line 98, 102. x \to \bm{x}$, line 98, 102, 108, 110, 145, 148, 150. `the following inequality' $\to$ `if the following inequality', line 250. `soft maximum', line 265. Supplementary Material: $x \to \bm{x}$, line 7. $f \to \bm{f}$, line 80. $\bm{x}$ should be $\bm {x}_a$, line 13. Missing full stop at the end of line 14, 15, 89-92, 95, 143. Missing comma, line 99. `forbitten', above line 251.

Relation to Prior Work: They show a detailed comparison with the most-commonly used soft-max function, exponential mechanism. And authors get new results on some common problems via a new way -- construct a piece-wise linear function, which is creative without repeating old methods.

Reproducibility: Yes

Additional Feedback: The feedback answers my question.


Review 2

Summary and Contributions: This paper studies approximation and smoothness properties of soft-max functions. The authors show that the commonly used exponential function has optimal tradeoff between expected additive approximation and smoothness measured with respect to Renyi divergence. However, once smoothness is measured with represent to l_p-norms, the exponential function is no longer the optimal one. The authors present a new the piecewise linear function which is the optimal one for norms, and besides the expected additive approximation, they also study its worst-case approximation. For multiplicative approximation, they present the power mechanism, which is optimal for Renyi Divergence. The paper also discusses many different applications of the new functions to mechanism design, sub modular optimisation, and deep learning.

Strengths: The paper is well-written, and the result seems to be quite important with many possible applications in many different topics. I think that the contribution of the paper will be appreciated by multiple sub-communities of NeurIPS. Post rebuttal: Thank you for your response.

Weaknesses: Approximate incentive compatibility is a rather weak concept in mechanism design, so I don't find this application very interesting. It would be nice to include proofs in the main text.

Correctness: Could not verify the claimed statements: no proofs are presented in the main text. However, the results seem plausible.

Clarity: The paper is quite well-written and easy to follow, which I really appreciate. The related literature could be discussed in more detail, and I think it would be nice to have a section with open problems/directions for future work.

Relation to Prior Work: The contribution of the paper is quite clear, but the related work could be discussed in more detail.

Reproducibility: No

Additional Feedback: -- Why is O(log(d)/delta) a "constant"? It depends on the input d, which can be as large as it wants, right? -- piece-wise --> piecewise -- line 77: take have been --> remove "take" -- line 86: usually, the unit simplex is denoted by Delta^{d-1}, not Delta_{d-1}


Review 3

Summary and Contributions: The paper proposed two soft-max functions: one denoted as PLSoftMax, for piece-wise linear soft max, and the other denoted as POW, for power mechanism. The paper investigated the theoretical properties of these two functions on the two aspects: (1) approximation and (2) smoothness, with intensive theoretical development. Further the paper demonstrated in empirical experiments how PLSoftMax can be used for solving differentially private submodular optimization problem on the data of DBLP computer scientist co-authorship networks; the result indicates that by comparing the exponential soft max functions, the proposed POW achieves better smoothness and approximation. Lastly, the paper claimed without empirical studies that (1) the PLSoftMax has good properties in the context of multi-class classification problems and (2) the PLSoftMax can be used to design an incentive compatible mechanism.

Strengths: (1) the paper claimed contributions to a wide spectrum of applications from mechanism design, differential privacy, and supervised learning (2) the paper provided comprehensive review of the theoretical development.

Weaknesses: (1) The motivation of why a better tradeoff between approximation and smoothness is clear; but the motivation of how the proposed soft-max functions are designed is unclear. Further, in addition to the theoretical proof, some more discussions or intuitions would be helpful for readers to understand better why empirically the proposed methods should be used. (2) No empirical support is provided for two of the three applications as claimed in the paper.

Correctness: Most of the claims are correct; the empirical methodology for sub-modular optimization is valid. The theoretical claims are not fully reviewed due to reviewer's time constraints. The claims on the applications on ML and mechanism design are not supported empirically.

Clarity: Yes, I enjoyed reading the paper a lot.

Relation to Prior Work: Yes.

Reproducibility: Yes

Additional Feedback: Other minor comments: (1) Line 135: some discussion would be nice to introduce why we want to switch perspective from (l_p, D_\inf)-Lipschitz to (l_p, l_q)-Lipschitz. Is it because (l_p, l_q)-Lipschitz is of particular interest to some applications or because that's the only setup where the approximation guarantee can be derived? (2) Line 346, the sentence is not well grounded by saying "useful in Machine Learning". More precise wording is needed, for example, "smoothness is preferred for gradient calculation in commonly adopted stochastic gradient descent algorithms". (3) In the appendix line 513, it'd be good to also report the standard deviation in addition the mean in the figure; this will help confirm the statistical significance visually.

[Author Response · NeurIPS 2020]

*We thank all reviewers for their reviews, for their constructive comments and suggestions and for appreciating our results. We apologize for the typos; all these minor points will be dealt with in the final version of our article.*

R#1: For Theorem 3.2, ... should be $c \geq \frac{\log d - 2}{2}$. Thank you for checking the proof of Theorem 3.2. Maybe there is a misunderstanding here but one of the main conclusions of Theorem 3.2 is that $c$ has to go to $\infty$ as $\delta \to 0$. Hence for small $\delta$ we cannot see how $c$ can be equal to $(\log d - 2)/2$. If you meant $c \geq \frac{\log d - 2}{2\delta}$ then our statement allows this.

In the Appendix B, ... are redundant. We agree that the dashed illustration may be redundant we will update our manuscript accordingly.

In line 173(Appendix), ... (this will not influence the result). Indeed this is a typo, thank you very much for finding this!

R#3: Approximate incentive ... in the main text. Approximate incentive compatibility is indeed a weak concept in many cases. At the same time though it has been a successful theoretical tool for designing mechanisms in settings where exact incentive compatibility is very difficult to achieve. Additionally the use of approximate incentive compatibility can vastly increase the theoretical performance of designed mechanisms [1].

We also commit on moving at least a proof sketch for our main theorems in the main part of the paper with more specific references to low level technical lemmas in the supplementary material.

The paper is quite .. for future work. Thank you for the suggestions! We will add more details in the related literature discussion and we will also add a conclusion section with open problems and future work.

R#5: (1) The motivation of why a better ... should be used. We commit on moving the main intuitions of the construction and our proofs to the main part of the paper to help the reader. We will also clarify the reasons for using our soft-max function in practice. We briefly name a few reason apart from the achievement of a better approximation - smoothness tradeoff:

- the worst-case approximation guarantee can be very appealing in many practical applications where bad outcomes have severe consequences even when they happen with low probability,
- our soft-max function has a simple piecewise linear form which allows the computation of both the function and its derivatives to be done very efficiently. This might be relevant in applications where computations and differentiations of a softmax function appear very frequently in some code.

(2) No empirical support is provided for two of the three applications as claimed in the paper. Thank you for the suggestion! In this work our target was to establish the necessary theoretical framework for this problem. We certainly agree that the next relevant step is to try our findings both on simulated and on real-world data where truncation occurs.

(1) Line 135: some ... can be derived? It is both actually. $(l_p, l_q)$-Lipschitz is of particular interest to some applications, for example in mechanism design the notion of smoothness related to incentive compatibility is the $(l_p, l_1)$-Lipschitzness. Additionally, to the best of our knowledge, in settings where worst-case approximation is very important then $(l_p, l_q)$-Lipschitzness is the only available notion, so far, of smoothness that gives non-trivial results.

(2) Line 346, the ... gradient descent algorithms". We will make sure to add more on that. The most important feature of our soft-max function that can be utilized in ML applications is the fact that favors sparse outcomes. This is formalized via the worst-case approximation guarantee of our soft-max functions. This sparsity in the output can be useful both because it is known to help generalization but also because it makes the execution of every step of SGD more efficient and hence can potentially have a big impact on the efficiency of training algorithms.

(3) In the appendix line 513, ... significance visually. Thank you for the suggestion! We will certainly add the standard deviation in the plots for the final version of the paper.

# References

[1] Frank McSherry and Kunal Talwar. Mechanism design via differential privacy. In *Foundations of Computer Science, 2007. FOCS'07. 48th Annual IEEE Symposium on*, pages 94–103. IEEE, 2007.


[Meta-Review · NeurIPS 2020]

This paper studies trade-offs between approximation quality and smoothness of "soft"-max functions. The natural exponential function has optimal tradeoff between expected additive approximation and smoothness measured with respect to Renyi divergence but suboptimal when measured via L_p norms. The authors present a new the piecewise linear function which is the optimal one for norms. The paper also discusses many different applications of the new functions to mechanism design, sub modular optimisation, and deep learning. The reviewers found this to be an well-written and thorough paper on an important problem of broad interest in machine learning. I recommend this paper for acceptance.